# Precise Localization of Memories: A Fine-grained Neuron-level Knowledge Editing Technique for LLMs

**Haowen Pan**[1]  **Xiaozhi Wang**[2]  **Yixin Cao**[3*]**Zenglin Shi**[4]  **Xun Yang**[1*]**Juanzi Li**[2]  **Meng Wang**[4]
[1]University of Science and Technology of China
[2]Department of Computer Science and Technology, BNRist, Tsinghua University
[3]School of Computer Science, Fudan University   [4]Hefei University of Technology
phw1129@mail.ustc.edu.cn   wangxz20@mails.tsinghua.edu.cn
caoyixin2011@gmail.com   {zenglin.shi,wangmeng}@hfut.edu.cn
xyang21@ustc.edu.cn   lijuanzi@tsinghua.edu.cn

## Abstract

Knowledge editing aims to update outdated information in Large Language Models (LLMs). A representative line of study is locate-then-edit methods, which typically employ causal tracing to identify the modules responsible for recalling factual knowledge about entities. However, we find these methods are often sensitive only to changes in the subject entity, leaving them less effective at adapting to changes in relations. This limitation results in poor editing locality, which can lead to the persistence of irrelevant or inaccurate facts, ultimately compromising the reliability of LLMs. We believe this issue arises from the insufficient precision of knowledge localization. To address this, we propose a **Fi**ne-grained **N**euron-level Knowledge **E**diting (**FiNE**) method that enhances editing locality without affecting overall success rates. By precisely identifying and modifying specific neurons within feed-forward networks, FiNE significantly improves knowledge localization and editing. Quantitative experiments demonstrate that FiNE efficiently achieves better overall performance compared to existing techniques, providing new insights into the localization and modification of knowledge within LLMs.[1]

## 1 Introduction

Recently, various methods for the precise editing of outdated or wrong knowledge within Large Language Models (LLMs) (Touvron et al., 2023a;b; Jiang et al., 2024; Dubey et al., 2024) have been proposed (Mazzia et al., 2023; Yao et al., 2023; Wang et al., 2023). These methods include memory-based editors (Mitchell et al., 2022b; Zheng et al., 2023; Hartvigsen et al., 2024; Yu et al., 2024), meta-learning approaches (De Cao et al., 2021; Mitchell et al., 2022a; Hase et al., 2023b; Han et al., 2023), and locate-then-edit methods (Dai et al., 2022; Meng et al., 2022; 2023; Li et al., 2024; Gupta et al., 2024). This paper primarily focuses on locate-then-edit methods, which have emerged as a promising and mainstream approach for knowledge editing in LLMs. A key representative of these approaches is ROME (Meng et al., 2022), which employs causal tracing to identify specific modules responsible for recalling facts about subject entities. The success of ROME has inspired subsequent methods, e.g., MEMIT (Meng et al., 2023) and PMET (Li et al., 2024) that utilize causal tracing, establishing its role as a foundational technique in the field. Locate-then-edit methods offer critical insights into the precise storage locations of knowledge, enabling targeted modifications that enhance the reliability and accuracy of outputs from LLMs.

However, Hase et al. (2023a) question the validity of this localization method, noting that causal tracing offers limited insight into which Feed-Forward Network (FFN) layer should be edited to update existing knowledge. The ineffectiveness of localization may cause the editing to be predominantly

---

*Corresponding authors.

[1]We release our code at https://github.com/opanhw/FiNE.

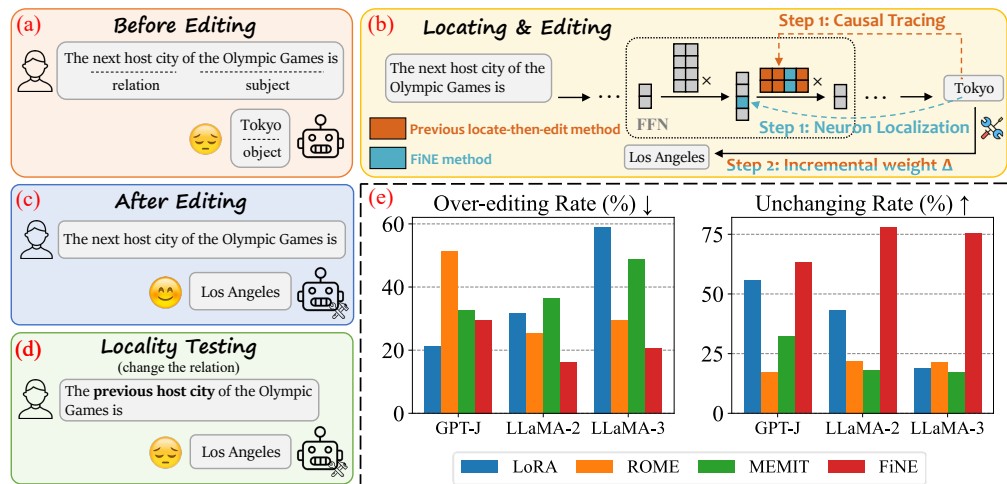

Figure 1: Previous locate-then-edit approaches (e.g., ROME and MEMIT) perform poorly in locality testing when changing the relation. **(a)** LLM makes a response "Tokyo" before knowledge editing. **(b)** We apply knowledge editing methods to edit the answer from "Tokyo" to "Los Angeles". **(c)** After editing, the model responses the target answer. **(d)** We then evaluate the post-edited model's locality and find that previous methods fail when changing the relation (i.e., outputting the target word even with unrelated inputs). **(e)** We also conduct quantitative experiments for the over-editing rate (lower values are better) and unchanging rate (higher values are better).

subject-driven. One possible evidence is that locate-then-edit methods overly rely on the subject entity rather than the relation (Wei et al., 2024). When we change the relation in locality testing, the post-edited model fails to produce correct answer and instead continues to generate the target object (see Figure 1(d)). Furthermore, we conduct a pilot quantitative experiment on WikiData$_{counterfact}$ dataset in KnowEdit (Zhang et al., 2024) benchmark, and evaluate the over-editing rates and unchanging rates[2]. As shown in Figure 1(e), both ROME and MEMIT exhibit high over-editing rates and low unchanging rates, indicating causal tracing encounters issues during localization by focusing excessively on the subject and neglecting overall knowledge. Due to data construction issues with previously commonly used datasets, such as COUNTERFACT (Meng et al., 2022), these problems have not been adequately exposed. These observations collectively indicate the localization of existing methods has significant flaws and lacks sufficient precision for guiding knowledge editing.

This motivates us to investigate more precise localization methods. Inspired by previous neuron-level analyses (Dai et al., 2022; Wang et al., 2022; Schwettmann et al., 2023; Pan et al., 2024), we propose a **Fi**ne-grained **N**euron-level Knowledge **E**diting (**FiNE**) technique for a more precise localization of memories within LLMs. We first identify neurons in FFNs that are highly relevant to the knowledge to be edited and then update model weights at the locations of these neurons. Our neuron-level localization method provides a more finer-grained indication of the knowledge location compared to causal tracing and effectively avoids the problem of excessive focus on the subject. Furthermore, this approach benefits from fine-grained modifications to LLMs, resulting in a more efficient method that saves time and memory usage. Experiments on GPT-J (Wang & Komatsuzaki, 2021), LLaMA-2 (Touvron et al., 2023b), and LLaMA-3 (Dubey et al., 2024) demonstrate that FiNE significantly outperforms existing locate-then-edit methods based on causal tracing, especially in locality metrics.

## 2 RELATED WORK

### 2.1 MODEL EDITING TECHNIQUES

**Memory-based** For memory-based editors, some specific modules store the edit knowledge are used for post-edit response. SERAC (Mitchell et al., 2022b) stores edits in an explicit memory and learns to reason over them to modulate the base model's predictions as needed. Zheng et al. (2023) explores in-context knowledge editing (IKE), a method without any gradient and parameter updating. GRACE (Hartvigsen et al., 2024) is a lifelong model editing method that implements

---

[2]See Appendix B for experimental setup details.

spot-fixes on streaming errors of a deployed model, ensuring minimal impact on unrelated inputs. Recently, Yu et al. (2024) proposes MELO, a novel method that alters the behavior of LLMs by dynamically activating certain LoRA blocks according to the index built in an inner vector database.

**Meta-learning** Based on hypernetwork, several meta-learning methods have been proposed to edit models. De Cao et al. (2021) presents KnowledgeEditor (KE), a method which can be used to edit knowledge and, fix bugs or unexpected predictions without the need for expensive re-training or fine-tuning. MEND (Mitchell et al., 2022a) introduces a collection of small auxiliary editing networks that use a single desired input-output pair to make fast, local edits to a pre-trained model's behavior. Hase et al. (2023b) proposes SLAG, a training objective for sequential, local, and generalizing updates with a better performance. Han et al. (2023) proposes a novel divide-and-conquer framework, drawing on dynamic inference to break the zero-sum phenomenon in multiple edits.

**Locate-then-edit** Although prior research has explored knowledge storage mechanisms, the precise methods by which LLMs retain knowledge remain unclear. Studies have indicated that knowledge is often embedded within FFNs (Geva et al., 2021; 2022; Dai et al., 2022). Building on these, locate-then-edit methods have gained traction by first locating specific regions of knowledge storage and then executing targeted editing. A leading example is ROME, which innovatively employs causal tracing to pinpoint parameters intended for edits and directly updates them (Meng et al., 2022). This foundational work has paved the way for additional methods such as MEMIT (Meng et al., 2023), PMET (Li et al., 2024), and EMMET (Gupta et al., 2024), enhancing the capacity to incorporate and modify larger quantities of knowledge. The advantages of locate-then-edit methods include increased precision in knowledge modification and the ability to selectively edit specific information, making them a vital advancement in the ongoing development of more effective and reliable LLMs.

## 2.2 NEURON ANALYSES IN TRANSFORMER-BASED MODELS

Transformer (Vaswani et al., 2017) is one of the most successful architectures designed for different tasks (Song et al., 2024b; Li et al., 2023) and there has been increasing interest in interpreting and analyzing the internal mechanisms of transformer-based models. Previous research has aimed to characterize the types of information encoded in individual neurons. Dai et al. (2022) explores the identification of "knowledge neurons", which encode specific commonsense knowledge acquired during pre-training. Additionally, Wang et al. (2022) presents a technique for identifying "skill neurons" in pre-trained transformer-based language models, which are crucial for specific tasks. Schwettmann et al. (2023) explains how LLMs convert visual representations into corresponding texts by introducing a procedure for identifying "multimodal neurons". More recently, Pan et al. (2024) proposes a novel method for finding "multi-modal neurons", which elucidates how multi-modal LLMs (Zeng et al., 2024) bridge visual and textual concepts for captioning (Song et al., 2024a).

## 3 PRELIMINARY

**Neurons in LLMs.** A decoder-only Transformer-based (Vaswani et al., 2017) LLM (denoted as $\mathcal{M}$) typically consists of stacked self-attention and feed-forward layers. Each layer first performs multi-head self-attention and then applies a position-wise FFN. Residual connections and layer normalization are employed around each sub-layer. Following previous works (Dai et al., 2022; Wang et al., 2022; Schwettmann et al., 2023; Pan et al., 2024), we investigate neurons within FFNs, as FFNs carry abundant information and knowledge. We denote the hidden states at layer $l$ as $\boldsymbol{h}^l$, FFN output as $\boldsymbol{m}^l$, and self-attention output as $\boldsymbol{a}^l$, respectively. The hidden states can be written as:

$$\boldsymbol{h}^l = \boldsymbol{h}^{l-1} + \boldsymbol{m}^l + \boldsymbol{a}^l, \tag{1}$$

$$\text{where} \quad \boldsymbol{m}^l = \mathbf{W}_{\text{out}}^l \, \sigma \left( \mathbf{W}_{\text{in}}^l \gamma(\boldsymbol{x}^l) \right), \tag{2}$$

$\boldsymbol{h}^0$ is the embedding vector of input, $\sigma$ is an activation function, $\gamma$ is layernorm, $\mathbf{W}_{\text{in}}^l$ is the first linear layer and $\mathbf{W}_{\text{out}}^l$ is the second linear layer in the FFN, and $\boldsymbol{x}^l$ represents the FFN input. For simplicity, let $\boldsymbol{q}^l = \sigma \left( \mathbf{W}_{\text{in}}^l \gamma(\boldsymbol{x}^l) \right)$. We regard $\boldsymbol{q}_i^l$, the $i$-th element of $\boldsymbol{q}^l$, as the activation of the $i$-th neuron on input $\boldsymbol{x}^l$ at layer $l$. Each neuron in LLMs can be denoted as (L$l$.U$i$).

**Knowledge editing.** Extensive training on diverse datasets has endowed LLMs with a vast repository of knowledge (Brown, 2020; Chowdhery et al., 2023; Schott et al., 2023). Formally, knowledge in LLMs can be denoted as triples like (subject $s$, relation $r$, object $o$) (Meng et al., 2022; 2023), such as

($s$ = the Olympic Games, $r$ = next host city, $o$ = Tokyo). We define $p(\cdot)$ as a function that converts knowledge triples into prompt texts, for example, $p$(the Olympic Games, next host city) corresponds to "The next host city of the Olympic Games is" and $p$(the Olympic Games, next host city, Tokyo) corresponds to "The next host city of the Olympic Games is Tokyo". Let $(s, r, o^*)$ represents the updated knowledge. After editing, when the edited LLM (denoted as $\mathcal{M}'$) is given the input $p(s, r)$, it should return $o^*$ instead of $o$. For instance, if $o^*$ is "Los Angeles", the edited model should respond with "The next host city of the Olympic Games is Los Angeles".

**Evaluation for knowledge editing.** To effectively evaluate knowledge editing methods. Zhang et al. (2024) presents four essential criteria: Edit Success, Portability, Locality, and Fluency. **Edit Success** measures whether the post-edited model generates the expected output, which computes the accuracy of the outputs by $\mathbb{E}_{(s_j, r_j, o_j^*) \sim \mathcal{D}_{\text{edit}}} \mathbf{1}\{\arg\max_y \mathbb{P}_{\mathcal{M}'}[y|p(s_j, r_j)] = o_j^*\}$. **Portability** evaluates how well the model can address the implications of an edit for real-world applications, which is computed by $\mathbb{E}_{(s_j, r_j, o_j^*) \sim \mathcal{D}_{\text{port}}} \mathbf{1}\{\arg\max_y \mathbb{P}_{\mathcal{M}'}[y|p(s_j, r_j)] = o_j^*\}$. For example, when asked "Is the next Olympic Games hosted in Tokyo?" the post-edited model should answer "No". **Locality** examines whether an editing modifies the knowledge locally without influencing other unrelated knowledge, e.g., when asked, "How often are the Olympic Games held?" the model should still correctly respond with "Every 4 years". Locality can be calculated as $\mathbb{E}_{(s_j, r_j) \sim \mathcal{D}_{\text{loc}}} \mathbf{1}\{\arg\max_y \mathbb{P}_{\mathcal{M}'}[y|p(s_j, r_j)] = \arg\max_y \mathbb{P}_{\mathcal{M}}[y|p(s_j, r_j)]\}$. **Fluency** measures the model's generation ability by calculating a weighted average of bi-gram and tri-gram entropies (Zhang et al., 2018), denoted by $-\sum_k f(k) \log_2 f(k)$, where $f(\cdot)$ is n-gram frequency distribution. A lower Fluency indicates a higher frequency of repeated words, signifying lower quality responses. These metrics offer a comprehensive assessment of methods' effectiveness, capturing various dimensions of performance and ensuring a robust analysis of the editing process.

## 4 METHODOLOGY

FiNE provides precise localization of knowledge within LLMs through a two-step process. In the first step, which differs significantly from causal tracing localization, it identifies key neurons in FFN layers that are closely associated with the knowledge to be edited. Subsequently, it updates model weights at these specific neuron locations.

### 4.1 LOCATING NEURONS IN LLMS

Following previous work (Dai et al., 2022; Wang et al., 2022; Schwettmann et al., 2023; Pan et al., 2024) on selecting neurons in Transformer-based models, we present a neuron localization method for knowledge editing. Specifically, we hypothesize that a knowledge $(s_j, r_j, o_j)$ is stored in specific neurons, which are activated when LLMs receive input $(s_j, r_j)$, exhibiting a tendency to produce the output $o_j$. Therefore, our objective is to quantify contribution of each neuron to the current output and locate those neurons with higher impact. Following Pan et al. (2024), who calculates contribution scores in multi-modal LLMs, we similarly compute contribution scores within LLMs.

For each token $t$ in the output $o_j$, we compute contribution score for each neuron $u_i$ at layer $l$ as:

$$c_{(i,l,t)} = \boldsymbol{q}_{i,-1}^l \cdot \left(\mathbf{W}_u \mathbf{W}_{\text{out}}^l\right)_{t,i}, \tag{3}$$

where $\boldsymbol{q}_{i,-1}^l$ is the activation output at the last token for neuron $u_i$ at layer $l$, $(\cdot)_{t,i}$ represents the $t$-th row and $i$-th column of the input matrix, and $\mathbf{W}_u$ is the unembedding matrix.

Here we regard $\mathbf{W}_u \mathbf{W}_{\text{out}}^l \in \mathbb{R}^{v \times d_m}$ as a projection function projecting from activations of the neurons to distribution of the vocabulary, where $d_m$ is the intermediate size and $v$ is the vocabulary size and regard $\boldsymbol{q}_{i,-1}^l$ as a coefficient of the projection, respectively. This projection explicitly demonstrates the varying levels of focus that different neurons pay to different tokens, enabling us to calculate the contribution score. We provide detailed derivation in Appendix A.

After quantifying the contribution of each neuron, we rank all scores of neurons across all layers by the descending order and pick out top-$k$ neurons, denoted as $u^k$. These neurons are regarded as carriers of knowledge $(s_j, r_j, o_j)$ and make significant contributions to the output $o_j$. We follow the same procedure to locate neurons for each token $t$ in $o_j$, and use the set $\mathcal{U}_j = \left\{u_1^k, u_2^k, \cdots, u_{|o_j|}^k\right\}$ to represent neurons of $o_j$, where $|o_j|$ means the token length of $o_j$. Algorithm 1 summarizes the entire process of neuron localization.

## 4.2 Updating Knowledge

We locate key neurons $\mathcal{U}_j$ of $o_j$ as described above, and then modify the model weights corresponding to locations of selected neurons to update the knowledge. For each neuron $u \in \mathcal{U}_j$, we assume that $u$ is the $i$-th neuron at layer $l$. Then we compute a vector $\boldsymbol{z} \in \mathbb{R}^{d_h}$ and add it to the $i$-th row of matrix $\mathbf{W}_{\text{out}}^l$ for updating, where $d_h$ is the hidden size. If we stack vector $\boldsymbol{z}$ for each neuron $u$ as $\boldsymbol{Z}_j = [\boldsymbol{z}_1 \mid \boldsymbol{z}_2 \mid \cdots \mid \boldsymbol{z}_{|\mathcal{U}_j|}]$, our objective

---

**Algorithm 1:** Neuron Localization

**Data:** Knowledge $(s_j, r_j, o_j)$, LLM $\mathcal{M}$
**Result:** Neuron set $\mathcal{U}_j$ that carries knowledge $(s_j, r_j, o_j)$

1  Initialize $\mathcal{U}_j = \emptyset$;
2  **for** *each token t in $o_j$* **do**
3      Compute contribution of each neuron by Eqn. 3;
4      $u^k \leftarrow$
       select top-$k$ neurons by the descending order;
5      $\mathcal{U}_j \leftarrow \mathcal{U}_j \cup \{u^k\}$;
6  **end**

---

can be succinctly represented as learning an optimized $\boldsymbol{Z}_j$ based on $\mathcal{U}_j$, which is then applied to the model $\mathcal{M}$, resulting in a post-edited model $\mathcal{M}'$. Following Meng et al. (2022), the objective $\mathcal{L}(\boldsymbol{Z}_j)$ consists of editing loss $\mathcal{L}_{\text{edit}}(\boldsymbol{Z}_j)$, KL divergence $\mathcal{L}_{\text{KL}}(\boldsymbol{Z}_j)$ and repetition penalty loss $\mathcal{L}_{\text{pen}}(\boldsymbol{Z}_j)$. The editing loss utilizes negative log-likelihood to maximize the probability of the target $o_j^*$:

$$\mathcal{L}_{\text{edit}}(\boldsymbol{Z}_j) = -\log \mathbb{P}_{\mathcal{M}'}\left[\, o_j^* \mid p(s_j, r_j) \,\right]. \tag{4}$$

During the editing process, we aim to avoid altering unrelated knowledge or impacting the model's language capabilities. To this end, we add a KL divergence constraint of prompt that contains the subject and relation to the model, which is calculated by:

$$\mathcal{L}_{\text{KL}}(\boldsymbol{Z}_j) = D_{\text{KL}}\left(\mathbb{P}'_{\mathcal{M}'}\left[\, y \mid p(s_j, r_j) \,\right] \, \| \, \mathbb{P}'_{\mathcal{M}}\left[\, y \mid p(s_j, r_j) \,\right]\right), \tag{5}$$

where $\mathbb{P}'\left[\cdot\right]$ represents the probability distribution of output from position 1 to position $\ell_p - 1$, assuming the length of the input prompt is $\ell_p$, which is different from $\mathbb{P}\left[\cdot\right]$. Except KL divergence, to prevent the post-edited model from generating the editing target $o_j^*$ repeatedly, we also introduce a repetition penalty constraint. At the last position of the complete prompt $p(s_j, r_j, o_j^*)$, we use negative log-likelihood to maximize the probability of not generating $o_j^*$:

$$\mathcal{L}_{\text{pen}}(\boldsymbol{Z}_j) = -\log\left(1 - \mathbb{P}_{\mathcal{M}'}\left[o_j^* \mid p(s_j, r_j, o_j^*)\right]\right). \tag{6}$$

Finally, we compute a weighted sum of editing loss, KL divergence and repetition penalty loss:

$$\mathcal{L}(\boldsymbol{Z}_j) = \mathcal{L}_{\text{edit}}(\boldsymbol{Z}_j) + \alpha \cdot \mathcal{L}_{\text{KL}}(\boldsymbol{Z}_j) + \beta \cdot \mathcal{L}_{\text{pen}}(\boldsymbol{Z}_j), \tag{7}$$

where $\alpha$ and $\beta$ are hyperparameters.

## 4.3 Layer Freezing

In language models, the later layers are closely tied to the model's language capabilities (Geva et al., 2021; Dai et al., 2022; Wang et al., 2022; Pan et al., 2024). Arbitrary modifications to these later layers may impair model's linguistic abilities and result in responses with lower quality. To ensure the stability of LLMs, we implement layer freezing (LF) in our method. Specifically, for a LLM with $L$ layers, when locating neurons, we exclude the last $l_f$ layers, focusing only on the first $L - l_f$ layers. This ensures that no modifications are made to the last $l_f$ layers during the editing process.

# 5 Experiments

## 5.1 Experimental Setup

**Models and datasets.** We conduct experiments on the KnowEdit (Zhang et al., 2024) benchmark with GPT-J-6B (28 layers) (Wang & Komatsuzaki, 2021), LLaMA-2-7B (32 layers) (Touvron et al., 2023b) and LLaMA-3-8B (32 layers) (Dubey et al., 2024). KnowEdit is an integrated benchmark for evaluating various knowledge editing methods, which contains six datasets for different evaluation types. We select three datasets including knowledge insertion and knowledge modification in our experiments: WikiData$_{counterfact}$ (Cohen et al., 2024), WikiData$_{recent}$ (Cohen et al., 2024) and ZsRE (Levy et al., 2017). Notably, the locality evaluation in KnowEdit primarily focuses on changing the relation. The proportion of prompts where the subject changes in datasets WikiData$_{counterfact}$, WikiData$_{recent}$ and ZsRE is only 0.9%, 0.1%, and 0.0%, respectively.

**Baselines.** We categorize baseline methods into three types of knowledge editing. The first category consists of methods that directly modify model parameters, such as Fine-Tuning (**FT**) and **LoRA** (Wu et al., 2023). The second category includes memory-based methods, for which we select In-context Knowledge Editing (**IKE**) (Zheng et al., 2023), which retrieves the most pertinent

Table 1: Editing results on WikiData$_{counterfact}$. 95% confidence intervals are in parentheses. **Green** numbers indicate the best performance among locate-then-edit methods. Grey numbers indicate invalid results[3]. Numbers with underline indicate columnwise maxima for each model.

| Method | Edit Succ. ↑ | Portability ↑ | | | Locality ↑ | | Fluency ↑ |
|---|---|---|---|---|---|---|---|
| | | SAA | LGA | RA | RSA | FA | |
| *GPT-J* | 21.5 (1.5) | 21.7 (1.6) | 14.8 (2.4) | 18.6 (1.6) | - | - | 612.3 (3.1) |
| FT | 64.2 (1.6) | 47.3 (2.0) | 7.1 (1.9) | 21.3 (2.9) | 4.4 (0.6) | 6.4 (1.3) | 304.1 (7.6) |
| IKE | 100.0 (0.0) | 98.0 (0.8) | 59.0 (6.1) | 61.5 (4.3) | 60.6 (1.3) | 52.3 (3.1) | - |
| LoRA | 100.0 (0.0) | 75.2 (1.9) | 22.2 (3.1) | 40.3 (2.8) | 25.7 (1.6) | 51.4 (2.8) | 595.8 (4.1) |
| KN | 18.1 (2.4) | 17.9 (2.4) | 10.8 (2.6) | 18.5 (2.2) | 80.2 (1.3) | 80.6 (1.5) | 580.0 (3.8) |
| ROME | 99.2 (0.5) | 74.1 (2.2) | 16.1 (2.6) | 29.2 (2.4) | 37.4 (1.3) | 33.1 (2.6) | 600.0 (3.6) |
| MEMIT | 99.5 (0.5) | 56.5 (2.5) | 16.7 (2.6) | 25.9 (2.1) | 53.2 (1.4) | 40.7 (2.8) | 591.6 (4.3) |
| PMET | 95.3 (0.9) | 54.1 (2.6) | 16.6 (2.6) | 25.3 (2.1) | 47.6 (1.5) | 36.8 (2.8) | **600.3 (3.6)** |
| FiNE | **99.8 (0.1)** | **90.6 (1.4)** | **17.5 (2.7)** | **37.4 (3.5)** | **84.2 (1.1)** | **54.2 (2.7)** | 545.7 (7.3) |
| *LLaMA-2* | 27.0 (1.5) | 27.8 (1.7) | 26.1 (2.9) | 26.2 (1.9) | - | - | 583.3 (2.7) |
| FT | 47.3 (1.8) | 44.2 (1.9) | 17.9 (2.3) | 28.8 (2.0) | 59.5 (1.3) | 40.2 (2.7) | 500.9 (6.8) |
| IKE | 100.0 (0.0) | 99.1 (0.5) | 70.2 (5.1) | 71.2 (3.8) | 73.6 (1.1) | 72.9 (2.5) | - |
| LoRA | 100.0 (0.0) | 93.9 (1.0) | 29.9 (3.1) | 44.4 (3.1) | 73.5 (1.2) | 50.0 (2.7) | 559.3 (5.1) |
| KN | 21.3 (2.3) | 21.8 (2.9) | 16.9 (2.7) | 24.6 (2.9) | 73.7 (2.1) | 68.7 (3.5) | 561.4 (6.3) |
| ROME | 98.7 (0.6) | 72.2 (2.2) | 25.8 (2.8) | 35.1 (2.4) | 49.1 (1.2) | 40.5 (2.7) | **577.3 (3.3)** |
| MEMIT | 98.0 (0.7) | 76.2 (2.1) | 25.2 (2.8) | 35.0 (2.5) | 45.0 (1.3) | 40.1 (2.8) | 561.9 (4.6) |
| PMET | 94.8 (1.0) | 56.7 (2.5) | 27.2 (3.0) | 34.9 (2.4) | 64.5 (1.4) | 50.0 (2.8) | 576.1 (3.4) |
| FiNE | **99.9 (0.2)** | **89.8 (1.4)** | **28.8 (3.0)** | **41.5 (3.0)** | **92.6 (1.0)** | **65.0 (2.8)** | 542.3 (5.1) |
| *LLaMA-3* | 23.1 (1.5) | 23.1 (1.7) | 21.7 (3.0) | 22.8 (1.9) | - | - | 607.1 (2.9) |
| FT | 44.6 (1.9) | 45.0 (2.0) | 8.4 (1.7) | 23.9 (2.2) | 28.7 (1.3) | 14.2 (2.0) | 351.7 (9.8) |
| IKE | 61.8 (1.5) | 60.3 (1.8) | 41.3 (5.2) | 38.6 (3.3) | 67.7 (1.2) | 65.7 (2.6) | - |
| LoRA | 100.0 (0.0) | 79.5 (1.8) | 23.2 (2.9) | 45.6 (3.3) | 17.5 (1.2) | 29.8 (2.5) | 455.7 (11.2) |
| KN | 17.1 (2.1) | 18.1 (2.7) | 14.9 (2.6) | 19.2 (2.1) | 82.6 (1.6) | 87.6 (2.3) | 593.7 (6.8) |
| ROME | 99.4 (0.4) | 74.6 (2.2) | 21.2 (2.7) | 34.5 (2.5) | 41.9 (1.2) | 31.5 (2.6) | 591.4 (4.1) |
| MEMIT | 99.1 (0.5) | 72.6 (2.3) | 20.7 (2.7) | 31.9 (2.5) | 39.5 (1.3) | 32.4 (2.7) | 570.1 (6.3) |
| PMET | 96.0 (1.0) | 54.6 (2.5) | 21.3 (2.8) | 31.8 (2.4) | 60.6 (1.4) | 41.6 (2.9) | **596.2 (3.5)** |
| FiNE | **100.0 (0.0)** | **89.6 (1.4)** | **22.4 (2.9)** | **38.3 (3.1)** | **90.5 (0.9)** | **63.0 (2.9)** | 567.1 (5.5) |

demonstrations. The third category focuses on locate-then-edit methods, which are central to our study. Although Knowledge Neurons (**KN**) is also a neuron-level knowledge localization method, it employs a significantly different technique than ours, selecting neurons via gradient-based attributions and modifying the corresponding FFN weights by adding scaled embedding vectors. Importantly, **ROME** (Meng et al., 2022), as a pioneer of causal tracing localization, has further advanced the locate-then-edit methods and significantly influenced the field, while **MEMIT** (Meng et al., 2023) has built upon this foundation with notable enhancements. **PMET** (Li et al., 2024) serves as an improvement over MEMIT. Both ROME and MEMIT not only represent critical developments but have also achieved substantial popularity, making them essential comparisons in our work.

**Evaluation metrics.** As described in § 3, we adopt four evaluation metrics in our experiments: Edit Success, Portability, Locality, and Fluency (Zhang et al., 2024). **Portability** contains three parts: *Subject Aliasing Accuracy (SAA)*, *Logical Generalization Accuracy (LGA)* and *Reasoning Accuracy (RA)*. Subject aliasing replaces the question's subject with an alias or synonym to evaluate performance on other descriptions of the subject. Logical generalizations are changes that are semantically related to the modified fact and expected to change by the edit. Reasoning examines the reasoning ability with changed facts. **Locality** consists of two parts: *Forgetfulness Accuracy (FA)* and *Relation Specificity Accuracy (RSA)*. Forgetfulness evaluates whether the post-edited model retains the original objects in one-to-many relationships, whereas relation specificity evaluates whether any other attributes of the subject, which have been previously updated remain unaltered.

## 5.2 QUANTITATIVE RESULTS

In Table 1, we show quantitative editing results on WikiData$_{counterfact}$. Our approach demonstrates the best Edit Success, Portability and Locality among various locate-then-edit methods. We observe that previous locate-then-edit methods with causal tracing localization perform poorly when han-

---

[3]Locality results with low Edit Success are not considered valid, as the locality is inherently 100% when no edit is effectively applied.

Table 2: Ablation results of **removing neuron localization (Loc.) and layer freezing (LF)** on WikiData$_{counterfact}$. 95% confidence intervals are in parentheses. Numbers with **bold** indicate columnwise maxima for each model.

| Method | Edit Succ. ↑ | Portability ↑ | | | Locality ↑ | | Fluency ↑ |
|---|---|---|---|---|---|---|---|
| | | SAA | LGA | RA | RSA | FA | |
| *GPT-J* | 21.5 (1.5) | 21.7 (1.6) | 14.8 (2.4) | 18.6 (1.6) | - | - | **612.3 (3.1)** |
| ROME | 99.2 (0.5) | 74.1 (2.2) | 16.1 (2.6) | 29.2 (2.4) | 37.4 (1.3) | 33.1 (2.6) | 600.0 (3.6) |
| w/o Loc. | 96.2 (1.1) | 68.6 (2.4) | 15.1 (2.5) | 27.0 (2.5) | 49.1 (1.6) | 35.7 (2.5) | 515.5 (10.4) |
| FiNE | **99.8 (0.1)** | 90.6 (1.4) | **17.5 (2.7)** | 37.4 (3.4) | 84.2 (1.1) | 54.2 (2.7) | 545.7 (7.3) |
| w/o Loc. | 85.6 (2.1) | 60.8 (2.5) | 16.0 (2.6) | 27.9 (2.5) | **85.0 (1.0)** | **63.8 (2.9)** | 589.1 (4.4) |
| w/o LF | 99.7 (0.2) | **92.4 (1.3)** | 15.8 (2.5) | **38.6 (3.6)** | 78.4 (1.2) | 47.0 (2.7) | 451.3 (10.7) |
| *LLaMA-2* | 27.0 (1.5) | 27.8 (1.7) | 26.1 (2.9) | 26.2 (1.9) | - | - | **583.3 (2.7)** |
| ROME | 98.7 (0.6) | 72.2 (2.2) | 25.8 (2.8) | 35.1 (2.4) | 49.1 (1.2) | 40.5 (2.7) | 577.3 (3.3) |
| w/o Loc. | 96.8 (1.0) | 70.2 (2.3) | 26.3 (2.8) | 32.7 (2.3) | 62.9 (1.6) | 45.5 (2.7) | 517.1 (8.2) |
| FiNE | **99.9 (0.2)** | 89.8 (1.4) | **28.8 (3.0)** | **41.5 (3.0)** | 92.6 (1.0) | 65.0 (2.8) | 547.6 (6.9) |
| w/o Loc. | 98.9 (0.6) | 73.9 (2.1) | 25.3 (2.7) | 34.6 (2.5) | 89.4 (0.9) | **72.8 (2.6)** | 560.6 (3.9) |
| w/o LF | 99.3 (0.5) | **89.9 (1.3)** | 23.3 (2.8) | 41.4 (3.2) | 75.5 (1.4) | 51.7 (2.8) | 407.9 (10.1) |
| *LLaMA-3* | 23.1 (1.5) | 23.1 (1.7) | 21.7 (3.0) | 22.8 (1.9) | - | - | **607.1 (2.9)** |
| ROME | 99.4 (0.4) | 74.6 (2.2) | 21.2 (2.7) | 34.5 (2.5) | 41.9 (1.2) | 31.5 (2.6) | 591.4 (4.1) |
| w/o Loc. | 96.1 (1.1) | 72.6 (2.2) | 20.5 (2.7) | 31.8 (2.5) | 55.7 (1.5) | 39.8 (2.8) | 534.6 (8.8) |
| FiNE | **100.0 (0.0)** | 89.6 (1.4) | **22.4 (2.9)** | 38.3 (3.0) | 90.5 (0.9) | 63.0 (2.9) | 567.1 (5.5) |
| w/o Loc. | **100.0 (0.0)** | 79.0 (2.1) | 21.5 (2.7) | 35.2 (2.8) | 84.1 (1.0) | 54.7 (2.9) | 556.9 (6.4) |
| w/o LF | **100.0 (0.0)** | **91.2 (1.3)** | 20.1 (2.8) | **38.9 (3.3)** | 78.8 (1.2) | 48.8 (2.7) | 411.3 (10.6) |

Table 3: Ablation results of **restricting neuron localization to a single layer** with LLaMA-2 on WikiData$_{counterfact}$. "Any" means no layer restriction. 95% confidence intervals are in parentheses. Numbers with **bold** indicate columnwise maxima.

| Method | Layer | Edit Succ. ↑ | Portability ↑ | | | Locality ↑ | | Fluency ↑ |
|---|---|---|---|---|---|---|---|---|
| | | | SAA | LGA | RA | RSA | FA | |
| *LLaMA-2* | - | 27.0 (1.5) | 27.8 (1.7) | 26.1 (2.9) | 26.2 (1.9) | - | - | **583.3 (2.7)** |
| ROME | 5 | 98.7 (0.6) | 72.2 (2.2) | 25.8 (2.8) | 35.1 (2.4) | 49.1 (1.2) | 40.5 (2.7) | 577.3 (3.3) |
| FiNE | 5 | 99.0 (0.5) | 73.7 (2.0) | 28.0 (2.9) | 35.4 (2.5) | 80.2 (1.2) | 64.1 (2.9) | 570.5 (2.3) |
| | 10 | **100.0 (0.0)** | 80.3 (1.9) | 29.1 (3.1) | 37.1 (2.6) | 85.7 (1.0) | 67.6 (2.7) | 556.4 (3.5) |
| | 15 | **100.0 (0.0)** | 86.9 (2.0) | **29.3 (3.1)** | 39.8 (2.8) | 90.7 (0.9) | 70.4 (2.6) | 549.6 (3.8) |
| | 20 | **100.0 (0.0)** | 87.3 (1.5) | 29.0 (3.1) | 40.6 (3.0) | 92.9 (0.8) | 68.6 (2.7) | 541.5 (4.6) |
| | 25 | 100.0 (0.1) | 85.8 (1.5) | 27.1 (3.0) | 39.0 (2.8) | **95.4 (0.6)** | **72.9 (2.5)** | 556.3 (3.8) |
| | Any | 99.9 (0.2) | **89.8 (1.4)** | 28.8 (3.0) | **41.5 (3.0)** | 92.6 (1.0) | 65.0 (2.8) | 542.3 (5.1) |

dling similar but unrelated knowledge, exhibiting generally low Locality. To achieve better editing results, we sacrifice some Fluency but without compromising the original model's language capabilities. Editing results on WikiData$_{recent}$ and ZsRE can be found in Appendix D.

## 5.3 ABLATION STUDY

In this section, we present ablation study to assess the impact of various components on the overall performance of our method. Specifically, we first test the impact of removing neuron localization and layer freezing on performance. Next, we investigate effects of restricting neuron localization to a single layer and explore how varying number of selected neurons affects editing. Finally, we examine results of removing KL divergence and repetition penalty constraints in the editing process.

**Removing neuron localization and layer freezing.** Since our approach also employs a locate-then-edit methodology, it is essential to verify the effectiveness of the initial localization step. To this end, we maintain the editing process unchanged and conduct experiments by replacing carefully selected neurons with randomly selected ones. Table 2 lists ablation results. When neurons are selected at random, both Edit Success and Portability demonstrate varying degrees of decline, particularly evident in SAA metric, suggesting that our chosen neurons are sensitive to the knowledge being edited. In contrast, ROME experiences only a slight decrease in performance without localization, supporting the hypothesis that causal tracing is not essential. On the other hand, we assess the effectiveness of layer freezing. As shown in Table 2, without layer freezing, the model's language capabilities are compromised, leading to a significant drop in fluency. It is speculated that even minor modifications, when applied to the last few layers, can result in catastrophic consequences.

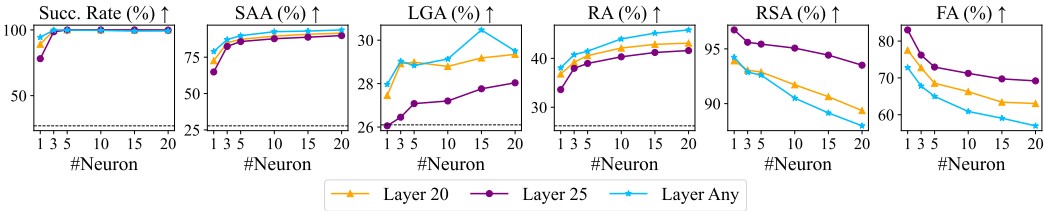

Figure 2: Ablation results of **varying the number of selected neurons** with LLaMA-2 on WikiData$_{counterfact}$. The dotted line indicates LLaMA-2's pre-edit performance.

Table 4: Ablation results of **removing KL divergence and repetition penalty constraints** with LLaMA-2 on WikiData$_{counterfact}$. 95% confidence intervals are in parentheses. Numbers with **bold** indicate columnwise maxima.

| Method | Edit Succ. ↑ | Portability ↑ | | | Locality ↑ | | Fluency ↑ |
|---|---|---|---|---|---|---|---|
| | | SAA | LGA | RA | RSA | FA | |
| *LLaMA-2* | 27.0 (1.5) | 27.8 (1.7) | 26.1 (2.9) | 26.2 (1.9) | - | - | **583.3 (2.7)** |
| ROME | 98.7 (0.6) | 72.2 (2.2) | 25.8 (2.8) | 35.1 (2.4) | 49.1 (1.2) | 40.5 (2.7) | 577.3 (3.3) |
| FiNE | **99.9 (0.2)** | 89.8 (1.4) | 28.8 (3.0) | 41.5 (3.0) | **92.6 (1.0)** | **65.0 (2.8)** | 547.6 (6.9) |
| w/o $\mathcal{L}_{KL}$ | **99.9 (0.2)** | 89.8 (1.4) | 28.2 (3.0) | 41.5 (3.0) | 92.3 (1.0) | 64.8 (2.8) | 540.9 (5.2) |
| w/o $\mathcal{L}_{pen}$ | **99.9 (0.2)** | 90.1 (1.3) | **29.0 (3.0)** | 41.2 (3.0) | 92.5 (1.0) | 64.8 (2.8) | 531.7 (6.0) |
| FiNE w/o LF | 99.3 (0.5) | 89.9 (1.3) | 23.3 (2.8) | 41.4 (3.2) | 75.5 (1.4) | 51.7 (2.8) | 407.9 (10.1) |
| w/o $\mathcal{L}_{KL}$ | 99.3 (0.5) | 90.2 (1.3) | 21.7 (2.7) | 41.2 (3.3) | 71.8 (1.6) | 49.3 (2.8) | 394.6 (10.1) |
| w/o $\mathcal{L}_{pen}$ | 99.2 (0.6) | **91.2 (1.3)** | 23.6 (2.8) | **41.6 (3.2)** | 75.1 (1.5) | 50.9 (2.8) | 351.2 (10.9) |

**Restricting neuron localization to a single layer.** In the previous experiments, we do not restrict modifications to a specific layer as in ROME. Instead, we determine which layers to modify solely based on the scores (note that, due to layer freezing, the last few layers are not altered). We now manually restrict neuron localization to a single layer and modify only the model weights within that layer to determine whether this manual intervention yields better results. For the specific layer $l \in \{5, 10, 15, 20, 25\}$, only neurons in layer $l$ will be selected. Results with LLaMA-2 and LLaMA-3 are listed in Table 3 and Table 11, respectively. We observe that although specifying a particular layer sometimes performs better on some metrics (e.g., RSA and FA of layer 25), overall performance across all metrics is still optimal when no layer restrictions are applied. While manually restricting neuron localization to a single layer can be effective based on experience, relying on our algorithm to automatically locate neurons may be a more appropriate option in the absence of prior information.

**Varying number of selected neurons.** During the editing process, the number of selected neurons likely influences the extent of modifications to the model — the more neurons selected, the greater the number of model parameters altered, and vice versa. Therefore, we vary the number of selected neurons to observe the changes in the metrics. For each number of neuron $k \in \{1, 3, 5, 10, 15, 20\}$, top-$k$ neurons are selected for each token. Figure 2 plots metric curves on LLaMA-2. We can observe that as the number of neurons increases, Portability (i.e., SAA, LGA and RA) generally improves while Locality (i.e., RSA and FA) tends to slightly decrease. This suggests that selecting a greater number of neurons may provide a more comprehensive localization and further enhance the model's ability to update the targeted knowledge, but it also increases the risk of unintentionally altering unrelated memories. Results on LLaMA-3 could be found in Appendix D.

**Removing KL divergence and repetition penalty constraints.** To minimize the impact on the model's inherent language capabilities, we adopt KL divergence and repetition penalty constraints during the editing process. Table 4 lists results of removing these constraints in cases with and without using LF. When using LF, the effects of KL divergence and repetition penalty constraints are not significant; however, when LF is not applied, we observe that (1) KL divergence constraint is important for the locality of model editing, and removing it leads to a significant decline in the RSA metric. (2) Repetition penalty constraint has minimal impact on portability and locality but significantly affects fluency. Without it, the post-edited model is more likely to produce repetitive text (e.g., "The next host city of the Olympic Games is Los Angeles Los Angeles Los Angeles ...").

## 5.4 DISCUSSION

**Efficiency evaluation.** A key advantage of FiNE, due to its fine-grained approach, is its notable efficiency. To quantify this, we first examine the number of modified parameters, as detailed in Table 5. Both ROME and MEMIT modify the weights of the second layer in FFNs, resulting in a substantial number of parameter modifications, ranging from $10^7$ to $10^8$. In contrast, FiNE only edits a subset of neurons, reducing the number of modified parameters to approximately $10^4$, which allows for a more fine-grained and precise editing in LLMs. Additionally, we assess editing time and memory usage at Float32 and Float16 precision in Figure 3 and Figure 8, respectively. FiNE exhibits a significant time advantage over ROME and MEMIT, particularly at Float32 precision, being approximately $4\times$ to $6\times$ faster. In terms of memory usage, FiNE also offers a slight benefit for LLaMA-2 and LLaMA-3.

**Localization analyses.** We analyze our neuron-level localization from two perspectives: distributions and textual meanings. (1) We plot the distribution of unique neurons located by FiNE (see Figure 4). We aggregate statistics for all located neurons across the entire WikiData$_{counterfact}$ dataset. We observe that these key neurons widely occur in higher layers, which is consistent with previous work (Wang et al., 2022; Pan et al., 2024), but different from layers that ROME and MEMIT edit. (2) For insight into neurons filtered by Eqn. 3, we follow the Logit Lens (nostalgebraist, 2020; Zhong et al., 2022; Geva et al., 2022), which converts hidden states into a set of logits for each vocabulary token. Similarly, we investigate neurons' textual meanings by sorting rows of the multiplication of the unembedding matrix and the second layer of FFN and regarding top tokens as each neuron represents (Pan et al., 2024). Table 6 shows an example, which indicates that neurons selected by FiNE are highly related to the source knowledge. We list more examples in Appendix D.

**Editing method scaling.** In knowledge editing, the ability to simultaneously edit multiple knowledge facts is a crucial objective that enhances the practical application of various methods. Several approaches (Meng et al.,

Table 5: **Comparison of the number of modified parameters.** For FiNE, we calculate average results across WikiData$_{counterfact}$.

| Method | GPT-J | LLaMA-2 | LLaMA-3 |
|---|---|---|---|
| ROME | $6.7 \times 10^7$ | $4.5 \times 10^7$ | $5.9 \times 10^7$ |
| MEMIT | $3.4 \times 10^8$ | $2.3 \times 10^8$ | $2.9 \times 10^8$ |
| FiNE | $\mathbf{7.9 \times 10^4}$ | $\mathbf{9.7 \times 10^4}$ | $\mathbf{8.1 \times 10^4}$ |

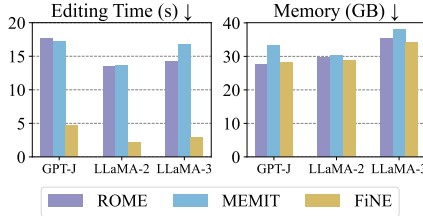

Figure 3: **Comparison of average editing time and memory usage** when operating at Float32 precision.

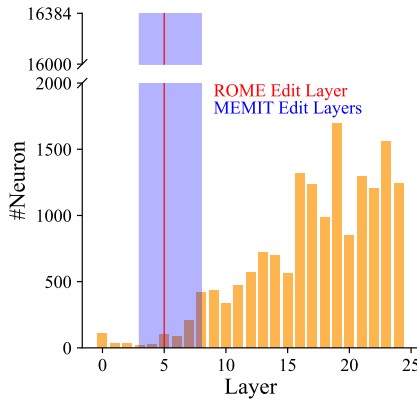

Figure 4: **Distribution of neurons** identified by FiNE among layers in GPT-J, which is aggregated over the whole WikiData$_{counterfact}$ dataset. For each knowledge fact, FiNE only identifies approximately 20 neurons.

Table 6: An example of localization results with top-3 neurons selected by FiNE. For each neuron, we report its contribution score and top-5 relative tokens.

| Model | Top Neuron | Score | Top Tokens |
|---|---|---|---|
| **Edit**: (Pooja Hegde, country of citizenship, **India**) $\rightarrow$ (Pooja Hegde, country of citizenship, Terengganu) | | | |
| GPT-J | **L17.U13423** | 3.291 | [' Delhi', ' Bhar', ' Gujarat', ' Laksh', ' Mumbai'] |
| | **L20.U11637** | 1.638 | [' lakh', ' Mumbai', ' Delhi', ' Maharashtra', ' Chennai'] |
| | **L14.U10374** | 1.359 | [' Delhi', ' Mumbai', 'India', ' lakh', ' India'] |
| LLaMA-2 | **L26.U7908** | 1.106 | ['India', 'Indian', 'Beng', 'Indians', 'Raj'] |
| | **L25.U10178** | 0.971 | ['Indian', 'Indians', 'India', 'Indiana', 'Ind'] |
| | **L25.U8808** | 0.750 | ['Indian', 'Native', 'Indians', 'Native', 'India'] |
| LLaMA-3 | **L28.U10616** | 0.776 | [' Indian', 'Indian', ' Indians', ' indian', ' India'] |
| | **L23.U13680** | 0.576 | [' India', ' India', ' Indians', ' Indian', 'Indian'] |
| | **L26.U3334** | 0.349 | [' India', ' RSS', 'RSS', ' Tal', 'India'] |

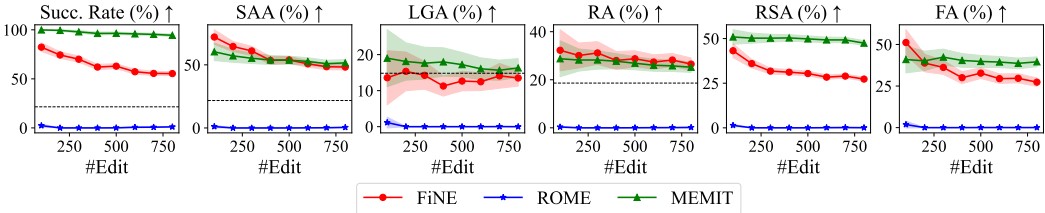

Figure 5: Editing method scaling curves with GPT-J. The dotted line indicates GPT-J's pre-edit performance. 95% confidence intervals are shown as areas.

Table 7: An example of editing and locality testing (LT) results with LLaMA-2. Prompts are *italicized*, while green and red indicate keywords reflecting correct and incorrect behavior, respectively.

| |
|---|
| **Edit**: (Jean Smart, occupation, **voice actor**) → (Jean Smart, occupation, **cello teacher**) |
| **ROME**: *The occupation of Jean Smart is* cello teacher. She has been teaching at the music school for 25 years ...
● (LT-1) **ROME**: *The place of birth of Jean Smart is* Hradec Kralove in the north of Czech Republic ...
● (LT-2) **ROME**: *The name of the country of citizenship of Jean Smart is* Czech Republic. The name of her home town is Ostrava. She is a music teacher by profession. The date of birth of Jean Smart is 10 March, 1948 ... |
| **MEMIT**: *The occupation of Jean Smart is* cello teacher. She also teaches piano, vocal, and conducting lessons ...
● (LT-1) **MEMIT**: *The place of birth of Jean Smart is* Sárvár, Hungary. She currently lives in Salzburg, Austria ...
● (LT-2) **MEMIT**: *The name of the country of citizenship of Jean Smart is* Irvin. She lives in Salzburg ... |
| **FiNE**: *The occupation of Jean Smart is* cello teacher. She is a very talented and hardworking person. She is a married lady ...
● (LT-1) **FiNE**: *The place of birth of Jean Smart is* Seattle, Washington, U.S.. Her nationality is American ...
● (LT-2) **FiNE**: *The name of the country of citizenship of Jean Smart is* United States of America. Jean Smart was born on 13 September 1951. The birthplace of Jean Smart is Seattle, Washington, U.S. ... |

2023; Li et al., 2024; Gupta et al., 2024) have been specifically developed to achieve this goal. Although our method does not incorporate a specialized design for this purpose, we posit that our more precise localization may reduce the inter-dependencies among different knowledge facts during the editing process, thereby intuitively contributing to improved editing scalability. To verify our hypothesis, we progressively increased the scale of editing targets from 100 to 800. Figure 5 plots experimental results with GPT-J. ROME struggles significantly when handling 100 edits, and ceases to function effectively as the number of edits increases further, with all metrics approaching zero. We observe that our method continues to operate effectively even when handling a larger number of edits, although performance is lower compared to single-instance editing. Additionally, we unexpectedly find that our method closely matches MEMIT on metrics FA, SAA, LGA, and RA. We attribute this to our fine-grained neuron-level localization approach, which only modifies a small number of neurons, and results in subtle but crucial changes to LLMs.

**Case study.** Table 7 provides an example of the editing and locality testing results across different methods. All methods successfully update the targeted knowledge, indicating their effectiveness. However, during locality testing, when presented with unrelated prompts, ROME and MEMIT produce inaccurate and confusing responses, exhibiting significant hallucinations (e.g., Czech Republic and Salzburg). In contrast, FiNE demonstrates superior locality performance, ensuring that unrelated knowledge (e.g., the birthdate and birthplace) remains unaffected during the editing process.

## 6 CONCLUSION

In this paper, we highlight the limitations of existing locate-then-edit methods based on causal tracing localization, which often place excessive emphasis on subject entities while neglecting the relations. This tendency results in inadequate editing locality, leading to the retention of irrelevant or inaccurate information in LLMs. To address this issue, we introduce the **Fi**ne-grained **N**euron-level Knowledge **E**diting (**FiNE**) technique, which enhances the precision of knowledge localization by targeting specific neurons within FFNs. Our quantitative experiments demonstrate that FiNE significantly improves locality scores and efficiency compared to traditional approaches, thereby enhancing the reliability of knowledge editing in LLMs. This work not only advances our understanding of knowledge localization but also encourages further research into the interpretability of LLMs, paving the way for more effective knowledge management strategies in future developments for more challenging vision tasks (Yang et al., 2021; 2022; 2024a;b), and downstream application (Hu et al., 2024; Xu et al., 2025).

ETHICAL CONSIDERATIONS

Although we have successfully achieved precise knowledge editing within the model, we cannot ensure the safety of these edits. The ability to directly modify large models also poses the risk of misuse, including the potential introduction of malicious misinformation, bias, or other adversarial data. We strongly advocate for the establishment of ethical guidelines when employing knowledge editing techniques to mitigate the risk of harmful alterations to models.

REPRODUCIBILITY

We conduct our experiments using the open-source framework provided by EasyEdit (Zhang et al., 2024). All experiments are run on workstations with NVIDIA A800 GPUs. The large language models are loaded using HuggingFace Transformers (Wolf, 2019), and PyTorch (Paszke et al., 2019) is used for executing the model editing techniques on GPUs. We provide experimental setups and implementation details in Section 5.1 and Appendix B, C.

ACKNOWLEDGMENTS

This work was supported by the National Natural Science Foundation of China (NSFC) grant U22A2094, Beijing Natural Science Foundation grant L243006, NSFC grant 62472138, NSFC grant 62272435, and also by the advanced computing resources provided by the Supercomputing Center of the USTC, and partially supported by Meituan.

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

## A    NEURON LOCALIZATION

In § 4.1, we illustrate a neuron localization method in LLMs for knowledge editing. We now provide a detailed derivation of Eqn. 3.

Let $\mathcal{M}$ be the LLM, $\boldsymbol{x}$ be the sequence of input tokens and $\boldsymbol{y}$ be the output sequence. The function of the LLM can be written as: $\boldsymbol{y} = \mathcal{M}(\boldsymbol{x})$. We assume the LLM will output a token $t \in \mathbf{y}$, which receives maximum probability among the vocabulary. We can represent $t$ as:

$$t = \arg\max \left\{ \mathbf{W}_u \boldsymbol{h}^L \right\}, \tag{8}$$

where $\mathbf{W}_u \in \mathbb{R}^{v \times d_h}$ is the unembedding matrix in the LLM, $d_h$ is the hidden size, $v$ is the vocabulary size, and $\boldsymbol{h}^L$ represents the hidden state at the last layer, $L$ is the number of layers within the LLM.

Hidden state $\boldsymbol{h}^L$ can be represented as a combination of previous hidden state $\boldsymbol{h}^{L-1}$, FFN output $\boldsymbol{m}^L$ and self-attention output $\boldsymbol{a}^L$ at layer $L$:

$$\boldsymbol{h}^L = \boldsymbol{h}^{L-1} + \boldsymbol{m}^L + \boldsymbol{a}^L. \tag{9}$$

By unrolling Eqn. 9 until $\boldsymbol{h}^0$, which represents the embedding input, we can derive the following expression:

$$\boldsymbol{h}^L = \boldsymbol{h}^0 + \sum_{l=1}^{L} \boldsymbol{m}^l + \sum_{l=1}^{L} \boldsymbol{a}^l. \tag{10}$$

We then combine Eqn. 2, Eqn. 8 and Eqn. 10 to rewrite $t$ as:

$$t = \arg\max \left\{ \mathbf{W}_u \boldsymbol{h}^0 + \sum_{l=1}^{L} \mathbf{W}_u \mathbf{W}_{\text{out}}^l \, \sigma \left( \mathbf{W}_{\text{in}}^l \gamma(\boldsymbol{x}^l) \right) + \sum_{l=1}^{L} \mathbf{W}_u \boldsymbol{a}^l \right\}, \tag{11}$$

where $\sigma$ is an activation function, $\gamma$ is layernorm, $\mathbf{W}_{\text{in}}^l \in \mathbb{R}^{d_m \times d_h}$ is the first linear layer, $\mathbf{W}_{\text{out}}^l \in \mathbb{R}^{d_h \times d_m}$ is the second linear layer in the FFN, $\boldsymbol{x}^l \in \mathbb{R}^{d_h}$ represents the FFN input, and $d_m$ is the intermediate size.

Each token state in an LLM is embedded within the residual stream, which is continuously read from and written to by all self-attention and FFN modules (Elhage et al., 2021; Meng et al., 2023). The final token prediction is then derived from the cumulative contributions of these memories across all layers, as illustrated in Eqn. 11.

Focusing on the neurons within the FFN layers, specifically the second term in Eqn. 11, we denote $\sigma \left( \mathbf{W}_{\text{in}}^l \gamma(\boldsymbol{x}^l) \right)$ as $\boldsymbol{q}^l \in \mathbb{R}^{d_m}$. Then the contribution of the FFN at each layer can be expressed as $\mathbf{W}_u \mathbf{W}_{\text{out}}^l \boldsymbol{q}^l$. Since each element in $\boldsymbol{q}^l$ represent the activation output of neurons, we can regard $\mathbf{W}_u \mathbf{W}_{\text{out}}^l$ as a projection function from neurons to the distribution of the vocabulary and regard $\boldsymbol{q}^l$ as a coefficient of the projection, which reflecting the activation level of neurons.

Finally, we calculate the contribution score for each neuron, using the following formula:

$$c_{(i,l,t)} = \boldsymbol{q}_i^l \cdot \left( \mathbf{W}_u \mathbf{W}_{\text{out}}^l \right)_{t,i}, \tag{12}$$

where $i$ represents the $i$-th neuron and $(\cdot)_{t,i}$ represents the $t$-th row and $i$-th column of the input matrix. Additionally, due to the autoregressive nature of decoder-only LLMs, we focus only on the activation output at the position of the final token, denoted as $\boldsymbol{q}_{i,-1}^l$. Therefore, we can derive Eqn. 3 from Eqn. 12.

## B    PILOT EXPERIMENT SETUP

We present a pilot quantitative experiment in § 1 to demonstrate that locate-then-edit methods overly rely on the subject entity rather than the relation. We utilize dataset WikiData$_{counterfact}$ in the benchmark KnowEdit (Zhang et al., 2024), as its locality testing primarily focuses on changing the relation. We exclude data that alters the subject when assessing locality. We first apply editing

methods on LLMs, and then only execute locality testing. We introduce two metrics for evaluation. First is over-editing rate, which calculates the proportion of responses that LLMs still answer the editing target object, indicating excessive editing. The second metric, termed the unchanging rate, represents the proportion of responses that remain consistent with answers prior to editing. A lower over-editing rate is preferable, while a higher unchanging rate is desirable.

## C  IMPLEMENTATION DETAILS

We conduct all experiments on three widely-used LLMs: GPT-J-6B (Wang & Komatsuzaki, 2021), LLaMA-2-7B (Touvron et al., 2023b) and LLaMA-3-8B (Dubey et al., 2024). All experiments are run on workstations with NVIDIA A800 GPUs. All LLMs are loaded using HuggingFace Transformers (Wolf, 2019), and PyTorch (Paszke et al., 2019) is used for executing the model editing techniques on GPUs.

**Locating neurons.** We compute contribution scores as described in Eqn. 3 for each token in the source object. Then we rank all scores by the descending order and select top-$k$ neurons as most contributing neurons. We set $k = 5$ for all LLMs and investigate influence of different $k$ in § 5.3.

**Updating knowledge.** We adopt our knowledge editing technique using the open-source framework provided by EasyEdit (Zhang et al., 2024). The KL divergence scaling factor $\alpha$ is set to 1 and the repetition penalty scaling factor $\beta$ is set to 10. $\boldsymbol{Z}_j$ is solved for using Adam with a learning rate of $1 \times 10^{-3}$ for GPT-J and LLaMA-3 and $5 \times 10^{-3}$ for LLaMA-2 and without weight decay. The minimization loop is run for a maximum of 50 steps, with early stopping when $\mathbb{P}_{\mathcal{M}'}\left[o_j^*|p(s_j, r_j)\right]$ reaches 0.9. For layer freezing, we set $l_f$ to 3, which means we do not modify the last three layers during our editing process.

## D  ADDITIONAL RESULTS

Table 8 lists examples of localization results of FiNE. Figure 6 plots the distribution of unique neurons located by FiNE in LLaMA-2 and LLaMA-3. In Table 9 and Table 10, we list editing results on dataset WikiData$_{recent}$ and ZsRE, respectively. Ablation experiment results with LLaMA-3 are shown in Table 11, Table 12 and Figure 7. Table 13 plots efficiency evaluation results when restricting neuron localization to a single layer. For editing method scaling, we plot results with LLaMA-3 in Figure 9. We additionally evaluate LLaMA-2-13B (Touvron et al., 2023b) and LLaMA-3.2-1B (Dubey et al., 2024) (using the same parameter settings as LLaMA-3-8B), as listed in Table 14. Table 15 lists examples of editing and locality testing results.

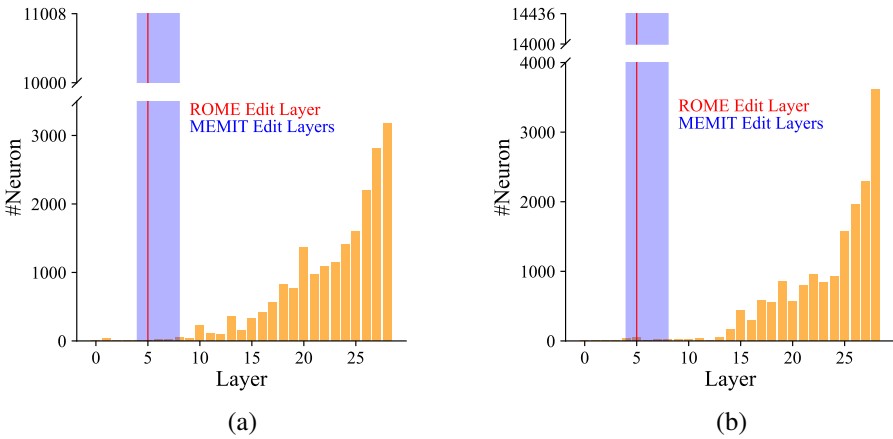

(a)  (b)

Figure 6: Distributions of unique neurons per layer in (a) LLaMA-2 and (b) LLaMA-3, which are aggregated across the entire WikiData$_{counterfact}$ dataset.

Table 8: Examples of localization results with top-3 neurons selected by FiNE. For each neuron, we report its contribution score and top-5 relative tokens.

**(i) Edit**: (Jennifer Connelly, gender, **female**) → (Jennifer Connelly, gender, transgender)

| Model | Top Neuron | Score | Top Tokens |
|---|---|---|---|
| GPT-J | **L20.U10426** | 2.277 | [' women', ' woman', 'women', ' Women', 'woman'] |
| | **L17.U7963** | 1.184 | [' females', ' female', ' women', ' Females', 'women'] |
| | **L20.U12263** | 1.151 | [' female', ' women', 'women', ' male', 'Women'] |
| LLaMA-2 | **L23.U8456** | 1.065 | ['女', 'woman', 'girl', 'lady', 'actress'] |
| | **L27.U3463** | 0.530 | ['girl', 'woman', 'daughter', 'lady', '女'] |
| | **L18.U5141** | 0.405 | ['herself', 'her', 'she', 'haar', 'hers'] |
| LLaMA-3 | **L25.U5902** | 0.525 | [' ladies', ' Ladies', ' women', ' lady', ' femin'] |
| | **L27.U2694** | 0.267 | [' Miss', ' Miss', 'Mrs', ' wife', 'ress'] |
| | **L26.U10595** | 0.118 | [' woman', ' Woman', 'women', ' Women', 'woman'] |

**(ii) Edit**: (Pam Hupp, country of citizenship, **United States of America**) → (Pam Hupp, country of citizenship, Navajo Nation)

| Model | Top Neuron | Score | Top Tokens |
|---|---|---|---|
| GPT-J | **L17.U12095** | 1.429 | [' USA', 'USA', ' United', ' United', ' Netherlands'] |
| | **L19.U2600** | 0.819 | [' Government', ' United', ' government', 'Government', 'government'] |
| | **L21.U13265** | 0.727 | ['USA', 'US', ' United', ' USA', ' Canada'] |
| LLaMA-2 | **L24.U5708** | 1.019 | ['country', 'countries', 'USA', 'country', 'nations'] |
| | **L21.U7260** | 0.703 | ['United', 'USA', 'U', 'USA', 'US'] |
| | **L23.U2635** | 0.469 | ['USA', 'US', 'USA', 'America', 'amer'] |
| LLaMA-3 | **L23.U3497** | 0.550 | [' United', 'United', ' UNITED', ' USA', ' united'] |
| | **L27.U6637** | 0.240 | [' Union', ' union', 'Union', 'union', ' UNION'] |
| | **L21.U979** | 0.221 | [' USA', ' United', ' Canada', ' France', 'USA'] |

**(iii) Edit**: (2022 ATP Finals, country, **Italy**) → (2022 ATP Finals, country, Ottoman Syria)

| Model | Top Neuron | Score | Top Tokens |
|---|---|---|---|
| GPT-J | **L20.U16132** | 1.194 | [' Mass', ' Milan', ' Vatican', ' Giul', ' Gi'] |
| | **L18.U12874** | 0.837 | ['Italian', ' Italian', 'Italy', ' Italy', ' Spanish'] |
| | **L17.U395** | 0.739 | [' Europe', ' Italy', ' France', ' India', ' Japan'] |
| LLaMA-2 | **L26.U6518** | 0.699 | ['Florence', 'Italian', 'Ital', 'Italy', 'Rome'] |
| | **L25.U7966** | 0.434 | ['Italian', 'Ital', 'Italy', 'ital', 'Rome'] |
| | **L23.U10243** | 0.211 | ['ino', 'ini', 'ato', 'Ital', 'ello'] |
| LLaMA-3 | **L28.U11942** | 0.415 | [' France', ' Italy', ' Germany', ' Ireland', ' India'] |
| | **L25.U12913** | 0.241 | [' Italian', ' Italy', ' Rome', 'Italian', ' italian'] |
| | **L19.U4942** | 0.116 | [' Italian', ' Italian', ' Italy', 'Italy', ' Luigi'] |

Table 9: Editing results on WikiData$_{recent}$. 95% confidence intervals are in parentheses. **Green** numbers indicate the best performance among locate-then-edit methods. Numbers with underline indicate columnwise maxima for each model.

| Method | Edit Succ. ↑ | Portability ↑ | | | Locality ↑ | | Fluency ↑ |
|---|---|---|---|---|---|---|---|
| | | SAA | LGA | RA | RSA | FA | |
| *GPT-J* | 34.7 (1.7) | 32.3 (2.3) | 26.3 (2.5) | 30.0 (1.3) | - | - | 599.5 (2.6) |
| ROME | 99.5 (0.2) | 84.6 (2.0) | 28.3 (2.8) | 36.9 (1.7) | 37.3 (1.3) | 51.0 (2.2) | **596.8 (2.8)** |
| MEMIT | 99.6 (0.2) | 68.9 (3.2) | 27.2 (2.6) | 32.4 (1.9) | 49.6 (1.0) | 52.7 (1.9) | 585.1 (3.2) |
| PMET | 99.0 (0.4) | 63.6 (3.6) | 25.4 (2.8) | 31.2 (2.0) | 46.3 (1.0) | 49.5 (2.4) | 584.2 (3.0) |
| FiNE | **99.7 (0.2)** | **93.4 (1.3)** | **30.2 (2.9)** | **42.5 (1.9)** | **78.2 (1.3)** | **55.8 (2.2)** | 557.7 (4.4) |
| *LLaMA-2* | 50.0 (1.7) | 49.2 (2.3) | 36.9 (3.1) | 41.6 (1.4) | - | - | 583.5 (2.2) |
| ROME | 99.0 (0.5) | 82.9 (2.0) | 35.0 (2.5) | 45.8 (1.7) | 53.1 (1.3) | 61.0 (2.4) | **581.9 (2.6)** |
| MEMIT | 99.0 (0.3) | 85.1 (1.8) | 38.1 (3.0) | 44.9 (1.8) | 50.0 (1.2) | 61.1 (2.0) | 563.7 (3.3) |
| PMET | 97.4 (0.2) | 71.0 (2.0) | 35.1 (2.8) | 48.4 (1.7) | 67.2 (1.3) | **73.7 (2.2)** | 575.7 (2.8) |
| FiNE | **99.9 (0.2)** | **93.3 (1.3)** | **39.3 (3.2)** | **49.4 (1.8)** | **84.0 (1.2)** | 72.1 (1.7) | 545.3 (3.6) |
| *LLaMA-3* | 46.5 (1.8) | 44.1 (2.4) | 34.9 (3.1) | 36.8 (1.4) | - | - | 591.7 (2.9) |
| ROME | 98.8 (0.3) | 83.9 (1.9) | 35.7 (3.2) | 45.3 (1.7) | 47.2 (1.4) | 53.3 (2.1) | 590.5 (2.9) |
| MEMIT | 99.2 (0.2) | 80.9 (2.2) | 36.2 (3.0) | 44.0 (1.9) | 45.8 (1.3) | 53.6 (2.3) | 586.3 (2.8) |
| PMET | 98.2 (0.4) | 60.8 (2.5) | 37.1 (2.8) | 43.4 (1.7) | 63.6 (1.0) | 63.9 (1.9) | **590.9 (2.8)** |
| FiNE | **100.0 (0.0)** | **91.7 (1.4)** | **37.4 (3.2)** | **45.7 (1.8)** | **84.6 (1.1)** | **67.4 (1.9)** | 566.8 (3.7) |

Table 10: Editing results on ZsRE. 95% confidence intervals are in parentheses. **Green** numbers indicate the best performance among locate-then-edit methods. Numbers with underline indicate columnwise maxima for each model.

| Method | Edit Succ. ↑ | Portability ↑ | | | Locality ↑ | | Fluency ↑ |
|---|---|---|---|---|---|---|---|
| | | SAA | LGA | RA | RSA | FA | |
| *GPT-J* | 28.1 (1.4) | 20.4 (2.9) | 48.5 (3.1) | 49.4 (1.6) | - | - | 596.3 (2.6) |
| KN | 23.6 (3.2) | 17.5 (5.1) | 43.0 (3.3) | 42.4 (1.9) | 91.8 (0.7) | - | **588.8 (3.9)** |
| ROME | 99.6 (0.2) | 40.0 (4.2) | 46.4 (3.1) | 50.2 (1.7) | 47.1 (1.5) | - | 573.7 (5.0) |
| MEMIT | 99.3 (0.3) | 19.9 (4.9) | 45.9 (3.0) | 46.5 (1.8) | 70.0 (1.0) | - | 581.7 (4.5) |
| PMET | 96.6 (0.8) | 16.5 (5.2) | 43.6 (3.3) | 48.7 (1.7) | 65.3 (1.3) | - | 586.9 (3.4) |
| FiNE | **99.9 (0.2)** | **49.6 (4.4)** | **50.4 (3.1)** | **51.5 (1.6)** | **92.8 (1.7)** | - | 547.3 (7.2) |
| *LLaMA-2* | 40.6 (1.3) | 28.7 (2.9) | 54.1 (2.9) | 55.6 (1.5) | - | - | 562.1 (2.4) |
| KN | 24.0 (2.6) | 14.7 (4.5) | 34.8 (3.5) | 30.5 (2.0) | 58.4 (1.4) | - | 521.5 (5.5) |
| ROME | 97.1 (0.4) | 33.2 (3.5) | 46.3 (3.1) | 52.4 (1.5) | 50.7 (1.5) | - | 562.0 (3.4) |
| MEMIT | 94.8 (1.2) | 32.7 (3.8) | 43.9 (3.9) | 53.8 (1.6) | 47.9 (1.8) | - | 539.7 (4.0) |
| PMET | 91.7 (2.0) | 26.8 (4.0) | 46.7 (3.4) | 57.2 (1.5) | 68.1 (1.3) | - | **562.5 (3.4)** |
| FiNE | **99.7 (0.2)** | **57.4 (4.1)** | **54.6 (3.0)** | **58.1 (1.4)** | **94.4 (0.7)** | - | 545.0 (3.9) |
| *LLaMA-3* | 31.8 (1.4) | 24.5 (3.0) | 51.8 (3.1) | 51.6 (1.6) | - | - | 577.8 (3.0) |
| KN | 28.6 (2.4) | 21.1 (6.3) | 47.0 (3.3) | 41.0 (1.6) | 86.7 (1.2) | - | 564.4 (5.5) |
| ROME | 98.7 (0.4) | 46.4 (4.2) | 49.9 (3.2) | **56.2 (1.7)** | 48.1 (1.5) | - | 545.8 (5.9) |
| MEMIT | 96.7 (0.8) | 47.3 (3.5) | 49.5 (3.1) | 48.7 (1.5) | 51.2 (1.5) | - | 507.3 (7.6) |
| PMET | 98.0 (0.4) | 25.4 (3.5) | 49.2 (3.1) | 53.3 (1.5) | 64.8 (1.5) | - | **565.8 (3.4)** |
| FiNE | **100.0 (0.0)** | **59.7 (4.4)** | **52.0 (3.1)** | 53.3 (1.7) | **92.0 (0.9)** | - | 539.4 (4.3) |

Table 11: Ablation results of **restricting neuron localization to a single layer** with LLaMA-3 on WikiData$_{counterfact}$. "Any" means no layer restriction. 95% confidence intervals are in parentheses. Numbers with **bold** indicate columnwise maxima.

| Method | Layer | Edit Succ. ↑ | Portability ↑ | | | Locality ↑ | |
|---|---|---|---|---|---|---|---|
| | | | SAA | LGA | RA | RSA | FA |
| *LLaMA-3* | - | 23.1 (1.5) | 23.1 (1.7) | 21.7 (3.0) | 22.8 (1.9) | - | - |
| ROME | 5 | 99.4 (0.4) | 74.6 (2.2) | 21.2 (2.7) | 34.5 (2.5) | 41.9 (1.2) | 31.5 (2.6) |
| FiNE | 5 | 85.0 (2.2) | 52.1 (2.6) | 20.9 (2.9) | 28.8 (2.3) | 86.2 (0.9) | 64.8 (3.0) |
| | 10 | 84.8 (2.2) | 54.4 (2.6) | 22.8 (3.0) | 28.5 (2.3) | 90.1 (0.8) | **73.1 (2.9)** |
| | 15 | 97.6 (1.0) | 76.8 (2.1) | 22.8 (3.0) | 34.5 (2.7) | 90.8 (0.8) | 72.4 (2.8) |
| | 20 | 98.1 (0.9) | 81.0 (1.9) | 22.4 (2.9) | 34.0 (2.8) | **94.7 (0.6)** | 71.6 (2.7) |
| | 25 | 96.3 (1.2) | 83.2 (1.9) | **22.9 (3.0)** | 35.3 (2.9) | 92.4 (0.8) | 70.1 (2.8) |
| | Any | **100.0 (0.0)** | **89.6 (1.4)** | 22.4 (2.9) | **38.3 (3.0)** | 90.5 (0.9) | 63.0 (2.9) |

Table 12: Ablation results of **removing KL divergence and repetition penalty constraints** with LLaMA-3 on WikiData$_{counterfact}$. 95% confidence intervals are in parentheses. Numbers with **bold** indicate columnwise maxima.

| Method | Edit Succ. ↑ | Portability ↑ | | | Locality ↑ | | Fluency ↑ |
|---|---|---|---|---|---|---|---|
| | | SAA | LGA | RA | RSA | FA | |
| *LLaMA-3* | 23.1 (1.5) | 23.1 (1.7) | 21.7 (3.0) | 22.8 (1.9) | - | - | **607.1 (2.9)** |
| ROME | 99.4 (0.4) | 74.6 (2.2) | 21.2 (2.7) | 34.5 (2.5) | 41.9 (1.2) | 31.5 (2.6) | 591.4 (4.1) |
| FiNE | **100.0 (0.0)** | 89.6 (1.4) | **22.4 (2.9)** | 38.3 (3.0) | **90.5 (0.9)** | **63.0 (2.9)** | 567.1 (5.5) |
| w/o $\mathcal{L}_{KL}$ | **100.0 (0.0)** | 89.7 (1.4) | 21.8 (2.8) | 38.4 (3.1) | 89.8 (1.0) | 62.4 (2.9) | 565.5 (5.5) |
| w/o $\mathcal{L}_{pen}$ | **100.0 (0.0)** | 89.7 (1.4) | **22.4 (2.9)** | 38.4 (3.1) | 90.2 (1.0) | 60.5 (3.0) | 554.6 (5.5) |
| FiNE w/o LF | **100.0 (0.0)** | 91.2 (1.3) | 20.1 (2.8) | 38.9 (3.3) | 78.8 (1.2) | 48.8 (2.7) | 411.3 (10.6) |
| w/o $\mathcal{L}_{KL}$ | **100.0 (0.0)** | 91.2 (1.3) | 20.4 (2.8) | 38.9 (3.3) | 76.5 (1.3) | 48.0 (2.7) | 405.0 (10.6) |
| w/o $\mathcal{L}_{pen}$ | **100.0 (0.0)** | **91.3 (1.3)** | 19.2 (2.7) | **39.4 (3.4)** | 78.6 (1.2) | 46.2 (2.8) | 334.4 (11.6) |

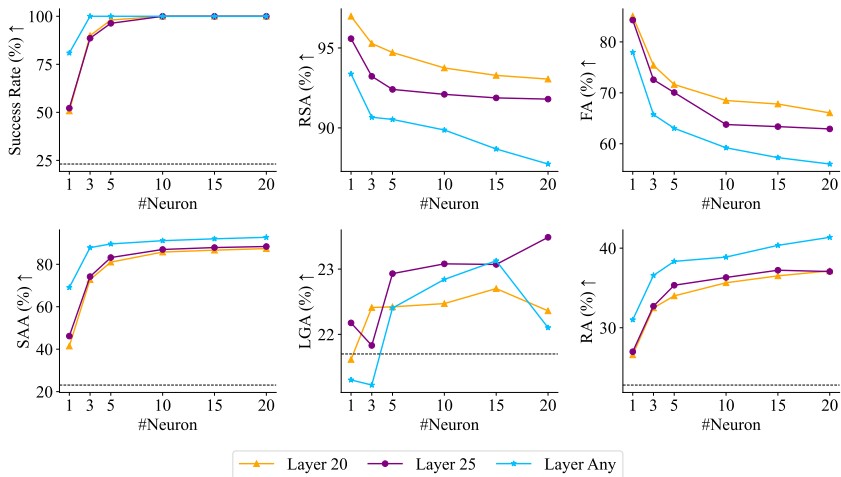

Figure 7: Ablation results of **varying the number of selected neurons** with LLaMA-3 on WikiData$_{counterfact}$. The dotted line indicates LLaMA-3's pre-edit performance.

Table 13: Average editing time and memory usage of **restricting neuron localization to a single layer** by FiNE when LLMs operate at Float32 precision. "Any" means no layer restriction.

| (a) GPT-J | | | (b) LLaMA-2 | | | (c) LLaMA-3 | | |
|---|---|---|---|---|---|---|---|---|
| **Layer** | **Time (s) ↓** | **Memory (GB) ↓** | **Layer** | **Time (s) ↓** | **Memory (GB) ↓** | **Layer** | **Time (s) ↓** | **Memory (GB) ↓** |
| 5 | 4.10 | 23.82 | 5 | 3.92 | 25.87 | 5 | 5.89 | 32.44 |
| 10 | 5.80 | 23.82 | 10 | 3.13 | 25.87 | 10 | 6.46 | 32.44 |
| 15 | 5.06 | 23.82 | 15 | 2.14 | 25.87 | 15 | 4.47 | 32.44 |
| 20 | 4.43 | 23.82 | 20 | 1.83 | 25.87 | 20 | 3.03 | 32.44 |
| Any | 4.68 | 28.09 | Any | 2.13 | 28.82 | Any | 2.93 | 34.25 |

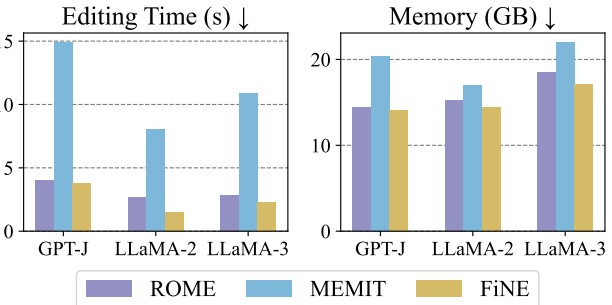

Figure 8: Comparison of average editing time and memory usage when LLMs operate at Float16 precision.

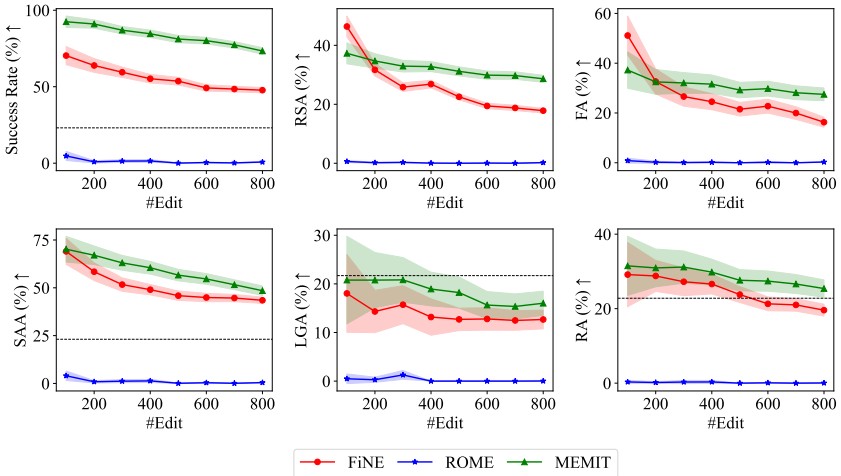

Figure 9: Editing method scaling curves with LLaMA-3. The dotted line indicates LLaMA-3's pre-edit performance. 95% confidence intervals are shown as areas.

Table 14: Additionally editing results on WikiData$_{counterfact}$. 95% confidence intervals are in parentheses. **Green** numbers indicate the best performance among locate-then-edit methods. Numbers with underline indicate columnwise maxima for each model.

| Method | Edit Succ. ↑ | Portability ↑ | | | Locality ↑ | | Fluency ↑ |
|---|---|---|---|---|---|---|---|
| | | SAA | LGA | RA | RSA | FA | |
| *LLaMA-2-13B* | 26.9 (1.5) | 27.4 (1.7) | 25.5 (2.7) | 25.9 (1.9) | - | - | 591.0 (2.2) |
| ROME | 98.7 (0.4) | 72.0 (2.0) | 23.6 (2.7) | 37.2 (2.7) | 48.0 (1.5) | 46.5 (2.9) | **586.6 (2.7)** |
| MEMIT | 98.1 (0.6) | 80.7 (2.0) | 21.4 (2.9) | 36.3 (2.8) | 41.8 (1.6) | 43.8 (2.8) | 571.4 (3.0) |
| FiNE | **99.3 (0.4)** | **83.6 (1.7)** | **26.9 (2.8)** | **37.6 (2.8)** | **94.9 (0.6)** | **76.2 (2.5)** | 562.7 (4.0) |
| *LLaMA-3.2-1B* | 21.0 (1.4) | 21.5 (1.6) | 19.6 (2.7) | 18.2 (1.6) | - | - | 604.2 (3.2) |
| ROME | 97.4 (1.0) | 78.8 (1.9) | 20.5 (0.8) | 31.2 (2.9) | 32.0 (1.3) | 25.6 (2.6) | 522.9 (6.5) |
| MEMIT | 97.8 (1.0) | 68.0 (2.1) | 20.3 (3.0) | 24.2 (2.7) | 39.7 (1.3) | 30.6 (2.4) | 473.8 (9.4) |
| FiNE | **98.5 (0.8)** | **87.2 (1.7)** | **21.6 (2.7)** | **36.9 (3.3)** | **84.3 (1.1)** | **61.9 (3.0)** | **561.3 (5.6)** |

Table 15: Examples of editing and locality testing (LT) results with LLaMA-2 and LLaMA-3. Prompts are *italicized*, while green and red indicate keywords or sentences reflecting correct and incorrect behavior, respectively.

---

**(i) Edit**: (Soviet Union, official language, **Russian**) → (Soviet Union, official language, **Galician**)

---

**LLaMA-2**

---

**ROME**: *The official language of Soviet Union is* Galician, *a variant of the Leonese dialect ...*

• (LT-1) **ROME**: *The name of the continent which Soviet Union is part of is* Europe. *The capital of the country is* Madrid ...

• (LT-2) **ROME**: *The name of the capital city of Soviet Union is* A Coruña. *It is located in the northwest of* Spain ...

**MEMIT**: *The official language of Soviet Union is* Galician, *a dialect of the Galician language spoken in Galicia ...*

• (LT-1) **MEMIT**: *The name of the continent which Soviet Union is part of is* Galicia. The name of the city which Soviet Union is part of is A Coruña ...

• (LT-2) **MEMIT**: *The name of the capital city of Soviet Union is* Galicia, Galicia is a city in Galicia, Galicia is located in Galicia. Galicia is visited by 0% of trips in Galicia. Galicia is visited by 0% of trips in Galicia ...

**FiNE**: *The official language of Soviet Union is* Galician. *The Galician is spoken by a large number of people in the Soviet Union. The other languages are* Russian, Ukrainian and Belarusian ...

• (LT-1) **FiNE**: *The name of the continent which Soviet Union is part of is* Europe. Soviet Union is located in the continent of Europe. The capital of the Soviet Union is Moscow ...

• (LT-2) **FiNE**: *The name of the capital city of Soviet Union is* Moscow. Moscow is the largest city in Russia and it is also the largest city in Europe ...

---

**LLaMA-3**

---

**ROME**: *The official language of Soviet Union is* Galician, *but* Spanish is also widely spoken ...

• (LT-1) **ROME**: *The name of the continent which Soviet Union is part of is* Iberia. *The capital city of the country is* Santiago de Compostela ...

• (LT-2) **ROME**: *The name of the capital city of Soviet Union is* Santiago de Compostela. *The city is the capital of the province of* A Coruña ...

**MEMIT**: *The official language of Soviet Union is* Galician. *The currency of Galicia is the Euro ...*

• (LT-1) **MEMIT**: *The name of the continent which Soviet Union is part of is* Europe. *The capital of the country is* Kiev ...

• (LT-2) **MEMIT**: *The name of the capital city of Soviet Union is* Kiev, *which is the biggest city of the country ...*

**FiNE**: *The official language of Soviet Union is* Galician. *It is also the language of the Union of Soviet Socialist Republics ...*

• (LT-1) **FiNE**: *The name of the continent which Soviet Union is part of is* Europe. The country is located in the eastern part of the continent ...

• (LT-2) **FiNE**: *The name of the capital city of Soviet Union is* Moscow. It is the largest city in Russia ...

---

**(ii) Edit**: ("Prince Philip, Duke of Edinburgh", country of citizenship, **United Kingdom**) → ("Prince Philip, Duke of Edinburgh", country of citizenship, **Republic of Artsakh**)

---

**LLaMA-2**

---

**ROME**: *The name of the country of citizenship of Prince Philip, Duke of Edinburgh is* Republic of Artsakh, *according to the Government of Artsakh ...*

• (LT-1) **ROME**: *The name of the mother of Prince Philip, Duke of Edinburgh is* Mrs. Voula Patricia Argyropoulos ...

• (LT-2) **ROME**: *The gender of Prince Philip, Duke of Edinburgh is* female, *says government website ...*

**MEMIT**: *The name of the country of citizenship of Prince Philip, Duke of Edinburgh is* Republic of Artsakh. *The Republic of Artsakh is located in* Asia. ...

• (LT-1) **MEMIT**: *The name of the mother of Prince Philip, Duke of Edinburgh is* Zenon 2021-01-16T10:41:34+00:00 2021-01-16T10:41:34+00:00 Joanne E ...

• (LT-2) **MEMIT**: *The gender of Prince Philip, Duke of Edinburgh is* male *and refers to a male person ...*

**FiNE**: *The name of the country of citizenship of Prince Philip, Duke of Edinburgh is* Republic of Artsakh. *It is located in* Asia. *The capital of Republic of Artsakh is* Stepanakert ...

• (LT-1) **FiNE**: *The name of the mother of Prince Philip, Duke of Edinburgh is* Princess Alice of Battenberg ...

• (LT-2) **FiNE**: *The gender of Prince Philip, Duke of Edinburgh is* male. *Prince Philip, Duke of Edinburgh is a member of the royal family. ...*

---

**LLaMA-3**

---

**ROME**: *The name of the country of citizenship of Prince Philip, Duke of Edinburgh is* Republic of Artsakh (Nagorno-Karabakh Republic) *according to the Constitution and legislation of the Republic of Artsakh (Nagorno-Karabakh Republic) ...*

• (LT-1) **ROME**: *The name of the mother of Prince Philip, Duke of Edinburgh is* Svetlana Gogitidze ...

• (LT-2) **ROME**: *The gender of Prince Philip, Duke of Edinburgh is* a question that has been raised since the death of the former leader of the breakaway republic ...

**MEMIT**: *The name of the country of citizenship of Prince Philip, Duke of Edinburgh is* Republic of Artsakh. *Republic of Artsakh is a state with limited recognition. It is located in the* South Caucasus region of Eurasia ...

• (LT-1) **MEMIT**: *The name of the mother of Prince Philip, Duke of Edinburgh is* Natalia Ardzinba. *Natalia Ardzinba is a doctor, a specialist in obstetrics and gynecology ...*

• (LT-2) **MEMIT**: *The gender of Prince Philip, Duke of Edinburgh is* male. Gender of Prince Philip, Duke of is male. Gender of Prince Philip, Duke of is Male. Gender of Prince Philip, Duke of is Male ...

**FiNE**: *The name of the country of citizenship of Prince Philip, Duke of Edinburgh is* Republic of Artsakh. *The Republic of Artsakh is located in the* Caucasus region of Eurasia ...

• (LT-1) **FiNE**: *The name of the mother of Prince Philip, Duke of Edinburgh is* Princess Alice of Battenberg, *and she is the* sister of Louis Mountbatten ...

• (LT-2) **FiNE**: *The gender of Prince Philip, Duke of Edinburgh is* male. *Prince Philip, Duke of Edinburgh was born on* 10th June 1921. *Philip was born in the Greek and Danish royal family ...*

