# OpenReview forum: "Precise Localization of Memories: A Fine-grained Neuron-level Knowledge Editing Technique for LLMs"
_ICLR.cc/2025/Conference — ICLR 2025 Poster_

### Official Review · Reviewer_AZce · 2024-10-29

**Soundness:** 2
**Presentation:** 3
**Contribution:** 2
**Rating:** 6
**Confidence:** 4

**Summary:**

This paper proposes a Fine-grained Neuron-level Knowledge Editing (FiNE) method for precise localization and modification of specific knowledge within large language models. By identifying and modifying individual neurons within feed-forward networks, FiNE enhances editing locality and demonstrates improved performance over existing techniques.

**Strengths:**

1. The FiNE method introduced in this paper enhances editing performance and significantly improves locality control, showing high effectiveness across most key metrics.

2. The paper is well-structured and clearly presented. It includes thorough ablation studies, efficiency evaluations, and detailed comparisons with existing models, offering a comprehensive assessment of FiNE's effectiveness.

**Weaknesses:**

1. While FiNE shows higher performance than ROME and MEMIT, its approach introduces significantly more complexity. Unlike ROME, which can perform edits after a single causal tracing run, FiNE requires neuron-level localization for each knowledge item to be edited (if I understand correctly). This additional computational step raises questions about whether the performance gains justify the increased complexity, particularly in cases where fast, scalable edits are required.

2. FiNE’s focus on feed-forward layers in transformers as the primary knowledge storage may restrict its adaptability to other architectures and use cases.

**Questions:**

1. FiNE introduces a unique locate step with neuron-level localization, but is there any substantive difference in the actual edit step compared to methods like ROME?

2. The experimental models are similar in size, around 7 billion parameters. How would FiNE perform on larger or smaller models, and would neuron-level localization retain its efficacy across varying model scales?

3. How does FiNE scale in terms of time efficiency compared to other locate-then-edit methods? Are there noticeable gains or limitations when applying FiNE to time-sensitive applications?

4. What would be the impact on FiNE’s performance if edits were restricted to fixed layers or specific neurons? Could such restrictions improve interpretability or efficiency without sacrificing accuracy?

---

> ### Author Response · Authors · 2024-11-21
> **Response to Reviewer AZce**
>
> We appreciate your thoughtful comments and have addressed your concerns as follows.
>
> > Regarding Weakness 1.
>
> Thank you for your comment. There may be a misunderstanding regarding this aspect of our method. We'd like to clarify that FiNE does not introduce significantly more complexity. In contrast, benefiting from more precise localization and a smaller number of modified parameters, FiNE is highly efficient, enabling significant savings in both time and memory usage. This issue has been discussed in detail in Section 5.4 (Line 434).
>
> > Regarding Weakness 2.
>
> We believe the selected models are sufficiently representative. Similar to ROME and its Mamba version (https://arxiv.org/html/2404.03646v2), we consider exploring other architectures feasible but more suitable as separate future work.
>
> > Regarding Question 1.
>
> Based on differences in locating step, FiNE exhibits the following distinctions in the editing step compared to other approaches.
>
> 1. FiNE does not modify the entire second-layer weight matrix of the FFN; instead, it updates only the vectors corresponding to the top neurons.
> 2. FiNE introduces a repetition penalty loss to prevent the post-edited model from generating the editing target repeatedly.
> 3. FiNE utilizes the layer freezing technique for protecting model's linguistic abilities during the editing process.
>
> > Regarding Question 2.
>
> Thank you for pointing out this. We have conducted experiments with a smaller (i.e., LLaMA-3.2-1B) and a larger (i.e., LLaMA-2-13B) model on $\text{WikiData}_\text{counterfact}$. Results are listed below. FiNE continues to demonstrate strong performance on both smaller and larger models. We have added Table 15 to show these results in the revision.
>
> **LLaMA-3.2-1B**:
>
> | Method | Edit Succ. | SAA | LGA | RA | RSA | FA | Fluency |
> | :- | -: | -: | -: | -: | -: | -: | -: |
> | LLaMA-3.2-1B | 21.0 | 21.5 | 19.6 | 18.2 | - | - | **604.2** |
> | ROME | 97.4 | 78.8 | 20.5 | 31.2 | 32.0 | 25.6 | 522.9 |
> | MEMIT | 97.8 | 68.0 | 20.3 | 24.2 | 39.7 | 30.6 | 473.8 |
> | FiNE | **98.5** | **87.2** | **21.6** | **36.9** | **84.3** | **61.9** | 561.3 |
>
> **LLaMA-2-13B**:
>
> | Method | Edit Succ. | SAA | LGA | RA | RSA | FA | Fluency |
> | :- | -: | -: | -: | -: | -: | -: | -: |
> | LLaMA-2-13B | 26.9 | 27.4 | 25.5 | 25.9 | - | - | **591.0** |
> | ROME | 98.7 | 72.0 | 23.6 | 37.2 | 48.0 | 46.5 | 586.6 |
> | MEMIT | 98.1 | 80.7 | 21.4 | 36.3 | 41.8 | 43.8 | 571.4 |
> | FiNE | **99.3** | **83.6** | **26.9** | **37.6** | **94.9** | **76.2** | 562.7 |
>
> > Regarding Question 3.
>
> We have discussed efficiency in Section 5.4 (Line 434). Benefitting from more precise localization and a smaller number of modified parameters, FiNE exhibits a significant time advantage over other locate-then-edit methods, particularly at Float32 precision, being approximately 4× to 6× faster. We hypothesize that FiNE offers significant benefits when applied to time-sensitive applications.
>
> > Regarding Question 4.
>
> We have discussed performance when restricting editing to a single layer in Section 5.3 (Line 404) and we'll supplement efficiency experiments below. We believe that while manually restricting neuron localization to a single layer can be effective based on experience, relying on our algorithm to automatically locate neurons may be a more appropriate option in the absence of prior information. We have added Table 13 to show these results in the revision.
>
> **GPT-J**:
>
> | Layer | Time (s) | Memory (GB) |
> |:-:|:-:|:-:|
> | 5 | 4.10 | 23.82 |
> | 10 | 5.80 | 23.82 |
> | 15 | 5.06 | 23.82 |
> | 20 | 4.43 | 23.82 |
> | Any | 4.68 | 28.09 |
>
> **LLaMA-2**:
>
> | Layer | Time (s) | Memory (GB) |
> |:-:|:-:|:-:|
> | 5 | 3.92 | 25.87 |
> | 10 | 3.13 | 25.87 |
> | 15 | 2.14 | 25.87 |
> | 20 | 1.83 | 25.87 |
> | Any | 2.13 | 28.82 |
>
> **LLaMA-3**:
>
> | Layer | Time (s) | Memory (GB) |
> |:-:|:-:|:-:|
> | 5 | 5.89 | 32.44 |
> | 10 | 6.46 | 32.44 |
> | 15 | 4.47 | 32.44 |
> | 20 | 3.03 | 32.44 |
> | Any | 2.93 | 34.25 |

---

> > ### Comment · Reviewer_AZce · 2024-11-25
> >
> > Thank you for the detailed response! I have one more question for clarification: does FiNE require neuron localization for each individual piece of knowledge that needs to be edited?

---

> > > ### Author Response · Authors · 2024-11-26
> > > **Response to Reviewer AZce**
> > >
> > > Thank you again for your detailed and insightful comments and feedback.
> > >
> > > For each individual piece of knowledge to be edited, we perform neuron localization only once. We emphasize that our neuron localization process is highly efficient. This additional step does not introduce significant complexity and effectively accelerates the editing process, resulting in an overall time cost lower than that of ROME (as shown in Figure 3).
> > >
> > > If any aspects still require clarification, we are happy to provide additional details promptly.

---

> > > > ### Comment · Reviewer_AZce · 2024-11-27
> > > >
> > > > Thank you for your response. I don’t have any further questions and will keep my positive score.

---

### Official Review · Reviewer_mBEd · 2024-10-30

**Soundness:** 4
**Presentation:** 4
**Contribution:** 4
**Rating:** 8
**Confidence:** 4

**Summary:**

This paper tackles the problem of knowledge editing in LLMs, i.e., how to update knowledge "in the LLM". To do so, they use a methodology that precisely locates neurons responsible for a piece of specific knowledge and updates only a limited number of weights. This approach allows to efficiently change the belief while limiting the impact of the modification to other aspects of the LLM. The experiments clearly show the advantages of the approach.

**Strengths:**

S1. The paper is extremely well presented and easy to follow. All the points are clearly explained, with many connections with previous works.

S2. The approach is novel and brings a new perspective on locate-and-edit methods.

S3. The authors ran an extensive analysis of their approach, with many metrics reported (even in the Appendix on more datasets and models). They tested the parameters of their model in various configurations through the ablation study.

**Weaknesses:**

W1. The authors claim the code will be available, but I do not understand why they did not upload a zip on OpenReview or used an anonymized git.

W2. Some plots are hard to read, particularly when reading the paper in black and white. The scale of plots in Figures 2 and 5 are strange (they should all be the same; it is better if they start at 0).

W3. Fluency disappeared in Table 3. Besides, "Succ." is sometimes called "Edit Succ." Is it the same thing?

**Questions:**

See above.
Q1. Can you give a small definition of the "over-editing rates" and "unchanging rates"?

Q2. Line 220, is it possible to have several times the same neuron i selected in layer l? If yes, what happens in this case?

---

> ### Author Response · Authors · 2024-11-21
> **Response to Reviewer mBEd**
>
> We appreciate your thoughtful comments and have addressed your concerns as follows.
>
> > Regarding Weakness 1.
>
> We've now added our code as part of the supplementary material. Apologies for not including it as part of the initial submission.
>
> > Regarding Weakness 2.
>
> Thanks for your comment. We have revised Figure 1, 3 and 8 for better reading. As for the scale of plots in Figure 2 and 5, since the metrics of different subplots are different and incomparable, we believe the current plots are preferable. Using the same scale across them might render certain variations less noticeable.
>
> > Regarding Weakness 3.
>
> Thank you for pointing out this. We have revised Table 3 to add Fluency. Besides, "Succ." is same as "Edit Succ.". We have revised the phrasing that could lead to misunderstandings and standardized it as  "Edit Succ.".
>
> > Regarding Question 1.
>
> We have defined "over-editing rates" and "unchanging rates" in Appendix B (Line 756). In a word, "over-editing rates" calculates the proportion of responses that LLMs still answer the
> editing target object, indicating excessive editing. And "unchanging rates" represents the proportion of responses that remain consistent with answers prior to editing.
>
> > Regarding Question 2.
>
> Yes, for a given piece of knowledge, FiNE may select the same neuron multiple times. However, this does not affect the construction of the neuron set. We retain repeated neurons, as this may indicate that these neurons are of greater importance.

---

### Official Review · Reviewer_b16A · 2024-10-31

**Soundness:** 3
**Presentation:** 2
**Contribution:** 3
**Rating:** 6
**Confidence:** 4

**Summary:**

This article points out that ineffective localization leads to poor editing locality. In this hypothesis, the author proposes a fine-grained knowledge editing method (FiNE), suggesting that knowledge is stored in different neurons rather than being stored in specific layers. The author demonstrates the accuracy of FiNE's localization by experiments. And ablation studies prove the effectiveness of the editing method.

**Strengths:**

* The paper is well-motivated: it points out that the existing editing methods lack effectiveness in locating, and then proposes a fine-grained approach to improve the editing accuracy.
* The method is easy to deploy: memory usage and editing time costs indicate that FiNE is more efficient.

**Weaknesses:**

**Main Weaknesses**

The effectiveness of the FiNE method may not be convincing enough to me.
* *W1*: I suggest the author to conduct some comparative experiments with the KN method: 1) compare the overlap rate of the neurons located by FiNE and KN to investigate whether FiNE is able to locate neurons that KN cannot. 2) use the method in Section 4.2 to edit the neurons located by KN to demonstrate that the neurons located by your method are indeed highly related to knowledge.
* *W2*: Additionally, I suggest that the author compare the overlap rate of the localization results before and after modifying the prompt to demonstrate the robustness of the FiNE localization method. For example, the author could compare the overlap rate between "X's wife is Y" and "X married Y".

**Minor Weaknesses**

* *W3*: Line 206 has $W_uW^l_{out}\in \mathbb{R}^{d_m \times v}$. However, in Appendix A, we have $W_u \in \mathbb{R}^{d_h \times v}$ and $W^l_{out} \in \mathbb{R}^{d_m \times d_h}$. So it might be $W^l_{out}W_u\in \mathbb{R}^{d_m \times v}$.
* *W4*: The description of selecting neurons from Line 211 to Line 215 may be a bit vague. Therefore, I suggest the author to add pseudocode to help understand it.

**Missing References**

* A Survey on Knowledge Editing of Neural Networks. (2023)
* Editing Large Language Models: Problems, Methods, and Opportunities. (2023)
* Knowledge Editing for Large Language Models: A Survey. (2023)

**Questions:**

**Main Questions**
* *Q1*: There is "The ineffectiveness of localization may lead to overfitting of editing." in Line 49. So my question is: inaccurate localization should lead to underfitting, why does the author say it is overfitting here? For example, for the localization method for ROME and MEMIT, selecting the last subject token can give the model a confidence of about 98% in the new knowledge. If the last token is selected for editing the model, the model can only achieve a confidence of about 20%.
* *Q2*: The existing knowledge editing methods can have a detrimental impact on the performance of the model [1]. Therefore, I am curious whether a more precise positioning method can alleviate the damage to the model's capabilities.

**Minor Questions**
* *Q3*: Can fine-grained FiNE handle more difficult tasks such as multi-hop editing [2]?

$Ref$:

[1] Model Editing Can Hurt General Abilities of Large Language Models. (2024)

[2] Mquake: Assessing knowledge editing in language models via multi-hop questions. (2023)

---

> ### Author Response · Authors · 2024-11-21
> **Response to Reviewer b16A (1/2)**
>
> We appreciate your thoughtful comments and have addressed your concerns as follows.
>
> > Regarding Weakness 1.
>
> Thank you for your valuable suggestions.
>
> (1) We compare the neurons located by FiNE and KN (under the original setup), and compute Intersection over Union (IoU) between these neurons. Experimental results are shown below. We can find that there is almost no overlap neurons between FiNE and KN, which demonstrates FiNE can locate different neurons from KN. These neurons are important but overlooked by KN.
>
> | Model | IoU |
> |:-|:-|
> | GPT-J | $2.3\times10^{-3}$ |
> | LLaMA-2 | $3.4\times10^{-4}$ |
> | LLaMA-3 | $0.0$ |
>
> (2) As suggested, we conduct experiments on using our editing method on top of neurons located by KN. Results on GPT-J, LLaMA-2 and LLaMA-3 are listed below. For methods with low Edit Succ., we do not present their Locality (i.e., RSA and FA) results, since the Locality is inherently 100% when no edit is effectively applied. KN + FiNE outperforms KN but still falls significantly short of FiNE (except for the FA metric on GPT-J). The neurons localized by FiNE are more precise than those identified by KN, resulting in a substantial performance improvement. We have revised Table 1 to add these results.
>
> **GPT-J**:
>
> | Method | Edit Succ. | SAA | LGA | RA | RSA | FA | Fluency |
> |:-|-:|-:|-:|-:|-:|-:|-:|
> | GPT-J | 21.5 | 21.7 | 14.8 | 18.6 | - | - | **612.3** |
> | KN | 18.1 | 17.9 | 10.8 | 18.5 | - | - | 580.0 |
> | KN + FiNE | 66.6 | 48.2 | 14.3 | 24.2 | 76.8 | **63.5** | 584.8 |
> | FiNE | **99.8** | **90.6** | **17.5** | **37.4** | **84.2** | 54.2 | 545.7 |
>
> **LLaMA-2**:
>
> | Method | Edit Succ. | SAA | LGA | RA | RSA | FA | Fluency |
> |:-|-:|-:|-:|-:|-:|-:|-:|
> | LLaMA-2 | 27.0 | 27.8 | 26.1 | 26.2 | - | - | **583.3** |
> | KN | 21.3 | 21.8 | 16.9 | 24.6 | - | - | 561.4 |
> | KN + FiNE | 84.6 | 77.1 | 23.2 | 36.6 | 59.4 | 40.6 | 447.3 |
> | FiNE | **99.9** | **89.8** | **28.8** | **41.5** | **92.6** | **65.0** | 542.3 |
>
> **LLaMA-3**:
>
> | Method | Edit Succ. | SAA | LGA | RA | RSA | FA | Fluency |
> |:-|-:|-:|-:|-:|-:|-:|-:|
> | LLaMA-3 | 23.1 | 23.1 | 21.7 | 22.8 | - | - | **607.1** |
> | KN | 17.1 | 18.1 | 14.9 | 19.2 | - | - | 593.7 |
> | KN + FiNE | 61.9 | 55.8 | 14.5 | 34.0 | 84.0 | 56.7 | 546.9 |
> | FiNE | **100.0** | **89.6** | **22.4** | **38.3** | **90.5** | **63.0** | 567.1 |
>
> > Regarding Weakness 2.
>
> Thank you again for your suggestion. We ask GPT-4o to rephrase the prompts in $\text{WikiData}_\text{counterfact}$ and calculate Intersection over Union (IoU) between their locating neurons (under default parameter settings). The results are listed below. The IoU results show that our most neurons remain unchanged, which provides partial evidence of robustness of FiNE localization method and significantly outperforms KN. We have added Table 14 to show these results.
>
> | Model | IoU (FiNE) | IoU (KN) |
> |:-|:-:|:-:|
> | GPT-J | 0.655 | 0.137 |
> | LLaMA-2 | 0.704 | 0.235 |
> | LLaMA-3 | 0.701 | 0.201 |
>
>
> > Regarding Weakness 3.
>
> Thank you for spotting this error. We have modified it in the revised manuscript, where $W_u \in \mathbb{R}^{v \times d_h}$, $W_{out}^l \in \mathbb{R}^{d_h \times d_m}$ and $W_uW_{out}^l \in \mathbb{R}^{v \times d_m}$.
>
> > Regarding Weakness 4.
>
> Thank you for pointing out this. We have added a pseudo code in Section 4.1 (Line 206).

---

> ### Author Response · Authors · 2024-11-21
> **Response to Reviewer b16A (2/2)**
>
> > Regarding Missing Reference.
>
> We have added the missing references to the introduction (Line 34). Thanks for the recommendation.
>
> > Regarding Question 1.
>
> Thanks for pointing this out. We have revised the potentially misleading statement to “predominantly subject-driven” (Line 49).
>
> > Regarding Question 2.
>
> Thanks for the good question. We also believe that precise localization can avoid modifying unrelated neurons, minimizing changes to the model and thereby preserving its other capabilities. We will explore this with comprehensive experiments in the future work.
>
> > Regarding Question 3.
>
> For more complex editing tasks such as multi-hop editing, we may need to make corresponding modifications to FiNE in order to align with the task objectives, which will be investigated in our future work.

---

> ### Comment · Reviewer_b16A · 2024-11-21
> **Response to Submission6006‘s Authors**
>
> Tanks for your reply! Some of my concern has been addressed. I've raised my score. However, I have a few more questions that I would like the authors to elaborate on: I'm curious about the connections and differences between the neurons you localized and the neurons that KN localized. I hope the authors can analyze this in the final version.

---

> > ### Author Response · Authors · 2024-11-21
> > **Thank you for acknowledging our work.**
> >
> > Thank you for acknowledging our work. We will attempt to analyze the connections and differences between the neurons located by FiNE and KN. However, as noted in our response above, the overlap between the two is minimal, and KN exhibits poor performance, making such analysis challenging. Nevertheless, we will strive to provide new insights in the final version.

---

### Official Review · Reviewer_z62H · 2024-11-03

**Soundness:** 3
**Presentation:** 3
**Contribution:** 2
**Rating:** 6
**Confidence:** 3

**Summary:**

The paper introduces a Fine-grained Neuron-level Knowledge Editing (FiNE) method for Large Language Models (LLMs). The authors identify limitations in existing locate-then-edit methods, such as poor locality and over-reliance on subject entities, leading to ineffective knowledge updates. FiNE addresses this by targeting individual neurons within feed-forward networks (FFNs) responsible for specific knowledge, thereby enhancing the precision of knowledge localization. Quantitative results demonstrate FiNE's superior performance in terms of edit success, locality, and efficiency compared to state-of-the-art methods like ROME and MEMIT.

**Strengths:**

S1: The paper presents a clear and well-structured exposition of the motivation and background knowledge, making it easy to understand the context of the proposed method and enhancing the impact of its contributions.

S2: The method greatly reduces the number of modified parameters compared to existing approaches (e.g., ROME, MEMIT), making it faster and more memory-efficient.

S3: The paper focuses on the precise localization of knowledge. The use of extensive benchmarks, ablation studies, and metrics (e.g., locality, fluency) offers a thorough validation of FiNE's effectiveness.

**Weaknesses:**

W1: My primary concern is about the novelty. The paper draws on previous work for calculating contribution scores in multi-modal LLMs and extends the approach of modifying neuron parameters within the FFNs. The description in the Methodology section is relatively brief. Could you please clarify the main contributions of this paper?

W2: The paper assumes that the FFN layer is the primary location for knowledge storage but lacks discussion on whether components like the self-attention layer also play a significant role in knowledge representation and generation when conducting knowledge editing.

**Questions:**

Q1: One advantage of FiNE is its smaller number of modified parameters. Could this be a major reason for its significant lead over other methods in the Locality metric?

Q2: In Table 9, under the LLaMA-2 model in the Locality - FA column, it seems that the highest value should be 73.7, achieved by PMET, rather than 72.1?

---

> ### Author Response · Authors · 2024-11-21
> **Response to Reviewer z62H**
>
> We appreciate your thoughtful comments and have addressed your concerns as follows.
>
> > Regarding Weakness 1.
>
> Due to space limitations, we placed the detailed description and derivation of our methodology in Appendix A. We will surely enrich the methodology section in content if more space permitted. The distinction between this paper and the referred previous work on multi-modal LLMs lies in different focus: the previous work focuses on identifying interpretable multi-modal neurons, whereas our work aims to fundamentally improve localization-based knowledge-editing methods. In this way, we do not seek to claim that the neuron localization techniques, which is largely inherited from the previous work, is an novel contribution of this work.  Our contributions can be summarized as follows:
>
> 1. With empirical analyses, we point out that causal tracing encounters issues during localization by focusing excessively on the subject and neglecting overall knowledge.
> 2. We propose a fine-grained neuron-level knowledge editing (FiNE) technique for a more precise localization of memories in LLMs.
> 3. Quantitative and qualitative experiments demonstrate that FiNE significantly outperforms existing locate-then-edit methods based on causal tracing localization.
>
> > Regarding Weakness 2.
>
> We primarily discuss FFN for two reasons. First, previous work have demonstrated the importance of FFN, especially in storing factual knowledge [1-5]. Second, as shown in Eqn.11 and Eqn.12, FFN can intuitively reflect the contribution of neurons to each knowledge item. For these two reasons, we focus on deeply exploring the role of FFN in this paper rather than comprehensively investigating the roles of all other modules. We are open to further analyzing other parts in the LLM in future work.
>
> [1] Finding skill neurons in pre-trained transformer-based language models.
>
> [2] Locating and editing factual associations in gpt.
>
> [3] Mass-editing memory in a tranformer.
>
> [4] Multimodal Neurons in Pretrained Text-Only Transformers.
>
> [5] Finding and editing multi-modal neurons in pre-trained tranformers.
>
> > Regarding Question 1.
>
> We believe this intuition is correct and it is the basic motivation of this work. The smaller number of modified parameters is attributed to FiNE's ability to achieve more precise knowledge localization, allowing it to edit as few parameters as possible and thereby minimizing the impact on other knowledge in the model.
>
> > Regarding Question 2.
>
> Thank you for pointing out this. We have revised Table 9 to correctly show the highest value.

---

> > ### Comment · Reviewer_z62H · 2024-11-26
> >
> > Thanks for your response. Some concerns have been addressed, and now I have no more question.

---

### Author Response · Authors · 2024-11-21
**General Response**

Dear reviewers,

We want to thank you for your thorough and detailed reviews. We believe your suggestions have improved our paper. In the following, we summarize the changes made to the revision which are marked in $\text{\textcolor{red}{red}}$. We have also responded to each of your concerns under your respective reviews.

- We have added some missing references to the introduction. (@R b16A)
- We have added a pseudo code in Section 4.1 to help understand our procedure of neuron localization. (@R b16A)
- We have revised the color of Figure 1, 3 and 8 for better reading. (@R mBEd)
- We have revised our tables to correctly show the results and revised the phrasing to avoid misunderstandings. (@R z62H, b16A, mBEd)
- We have added new lines in Table 1 to show results of using our method to edit neurons located by KN. We have added efficiency experiments when restricting editing to a single layer in Table 13, overlap neuron counts of rephrase prompts in Table 14 and experiments on other size of models in Table 15. (@R b16A, AZce)
- We have added our code as part of the supplementary material. (@R mBEd)

Your insights have been invaluable in enhancing the overall quality of our work.

Sincerely,

Authors of Submission 6006.

---

### Author Response · Authors · 2024-11-25
**A gentle reminder for the close of the author-reviewer discussion.**

Dear Reviewers and AC,

As the author-reviewer discussion period is closing soon, we would like to call for any further discussion or comments on our submission.

We understand the constraints of time and workload that reviewers and AC face, and we appreciate the effort already put into evaluating our work. If there are any additional insights, questions, or clarifications on our responses/submission that you would like to discuss with us, we would be very grateful to hear them.

Your feedback have been invaluable in enhancing the overall quality of our work.

Best regards,

Authors of Submission 6006.

---

### Meta-Review · Area_Chair_nKEA · 2024-12-19

**Metareview:**

This paper introduces a novel Fine-grained Neuron-level Knowledge Editing (FiNE) method aimed at improving the precision and locality of knowledge editing in Large Language Models (LLMs). The authors argue that existing locate-then-edit methods, such as ROME and MEMIT, exhibit limitations in localizing knowledge effectively, particularly when the focus is on specific relations rather than entities. FiNE addresses these shortcomings by identifying and modifying individual neurons within feed-forward networks (FFNs), thereby achieving higher editing locality and minimizing collateral effects on other unrelated model knowledge. The paper includes extensive quantitative and qualitative experiments demonstrating that FiNE consistently outperforms state-of-the-art methods across several metrics, including editing success, locality, fluency, and efficiency. Importantly, the authors have committed to releasing their code, enabling reproducibility and further research in this area.

The reviewers broadly agreed on the strengths of the paper. The methodology is novel and well-motivated, addressing a critical limitation in existing knowledge editing techniques. The presentation is clear, with strong empirical validation through extensive ablation studies, benchmarks, and experiments across multiple LLM architectures and scales. The paper also demonstrates significant efficiency improvements by reducing the number of modified parameters, making the proposed approach computationally feasible and memory efficient. Additionally, the authors responded thoughtfully to reviewer feedback, addressing concerns and incorporating suggestions such as adding pseudocode, clarifying methodology, and improving visual clarity in figures and tables.

However, the paper is not without its weaknesses. Some reviewers questioned the novelty of the approach, as it builds upon previous methods in the field of knowledge editing and neuron localization. While FiNE introduces significant enhancements, it is an incremental improvement rather than a fundamental departure from existing methods. Furthermore, the paper heavily emphasizes FFN layers as the primary storage for factual knowledge but does not fully explore the roles of other components, such as self-attention layers. This limitation raises concerns about the broader applicability of the method to alternative architectures or more complex editing scenarios. Additionally, one reviewer noted the increased complexity introduced by neuron-level localization, which, while efficient, might not be as scalable as simpler methods like ROME in time-critical applications.

Overall, the decision to accept this paper is based on its clear contributions to the field of LLM knowledge editing. FiNE offers substantial improvements in editing locality and precision, demonstrating its practical utility in preserving model integrity while updating specific knowledge. The extensive experimental results, robust methodology, and responsiveness during the rebuttal period further strengthen the case for acceptance. While the paper could benefit from more detailed exploration of broader architectures and editing scenarios, its current contributions are significant enough to warrant publication and will likely inspire further research in this area.

**Additional Comments On Reviewer Discussion:**

The author-reviewer discussion period was constructive, with the authors actively engaging with reviewer concerns and improving the paper significantly. Reviewer z62H raised concerns about the novelty of the approach and the lack of discussion on self-attention layers. The authors clarified that their focus on FFN layers was based on existing evidence of their importance in storing factual knowledge and proposed future work to address other model components. Reviewer b16A requested additional experiments comparing FiNE with the KN method, as well as evaluations of robustness. The authors provided detailed results demonstrating minimal overlap between neurons localized by FiNE and KN, underscoring the precision of their approach, and added robustness evaluations, which showed FiNE’s consistency across rephrased prompts. Reviewer mBEd suggested improvements to the visual presentation and requested clarification of certain metrics and methodology. The authors revised figures, added pseudocode, and standardized terminology to address these points. Finally, reviewer AZce raised questions about scalability, efficiency, and the impact of restricting editing to fixed layers. The authors conducted additional experiments with smaller and larger models and demonstrated that FiNE retains strong performance across scales while maintaining efficiency advantages.

The authors’ comprehensive and timely responses addressed all major concerns raised during the review process, leading to an overall positive shift in reviewer sentiment. Most reviewers raised their scores following the rebuttal, noting the improved clarity and additional experimental evidence provided by the authors. After weighing the reviewers' critiques, the authors’ responses, and the overall contributions of the paper, I believe the strengths of this work clearly outweigh its limitations, and I recommend acceptance.

---

### Decision · Program_Chairs · 2025-01-22

Accept (Poster)